# Native-Resolution Image Synthesis

**Zidong Wang**[1], **Lei Bai**[2,*], **Xiangyu Yue**[1], **Wanli Ouyang**[1,2], **Yiyuan Zhang**[1,2] [*]

[1]CUHK MMLab     [2]Shanghai AI Lab

wangzd2022@gmail.com, xyyue@cuhk.edu.hk, ouyangwanli@pjlab.org.cn
Project Page: https://wzdthu.github.io/NiT

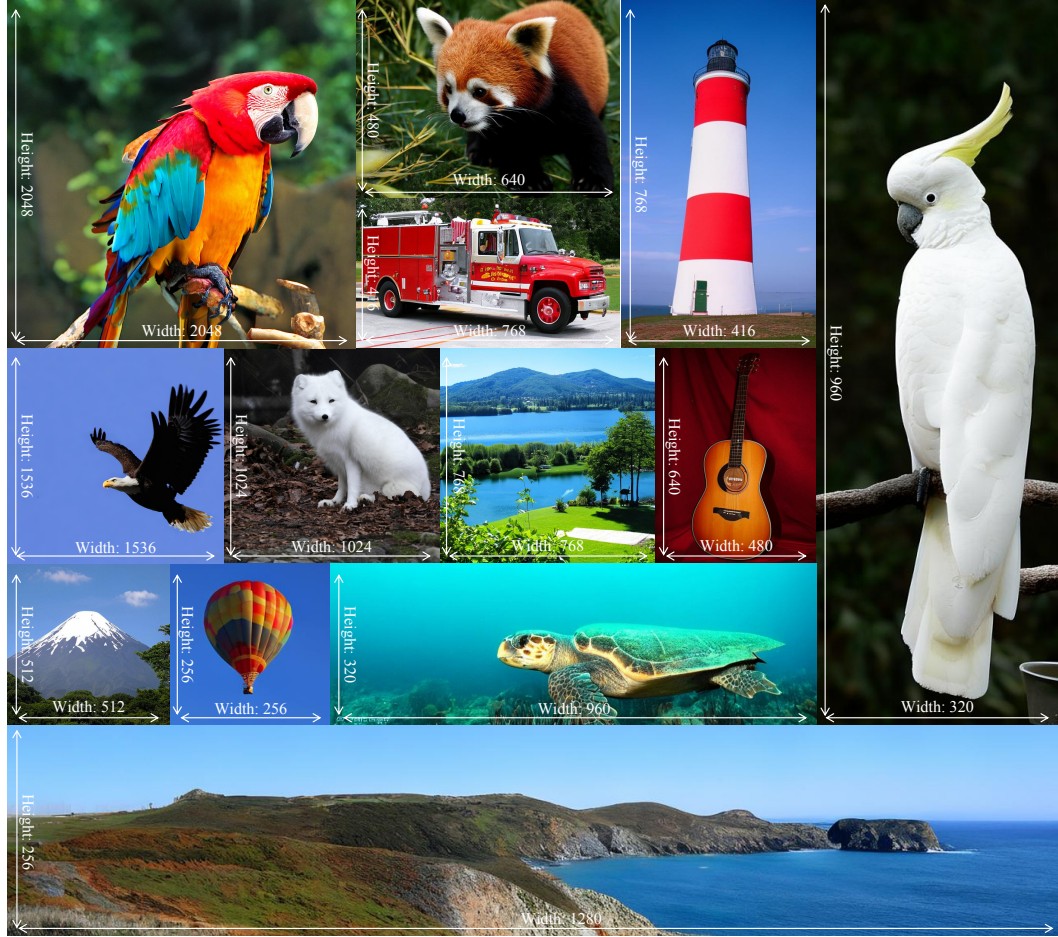

Figure 1: **Native-resolution image synthesis on *ImageNet*.** A single Native-resolution diffusion Transformer (NiT) model, trained on *ImageNet*, generates images across diverse, arbitrary resolutions and aspect ratios (examples shown from $256 \times 256$ to $2048 \times 2048$, and aspect ratios from $1:5$ to $3:1$). This capability extends far beyond conventional fixed-resolution, square-image generation (*e.g.*, $256 \times 256$), demonstrating strong generalization.

## Abstract

We introduce native-resolution image synthesis, a novel generative modeling paradigm that enables the synthesis of images at arbitrary resolutions and aspect ratios. This approach overcomes the limitations of conventional fixed-resolution, square-image methods by natively handling variable-length visual tokens—a core

---

[*]Corresponding authors: yiyuan@link.cuhk.edu.hk, bailei@pjlab.org.cn

39th Conference on Neural Information Processing Systems (NeurIPS 2025).

challenge for traditional techniques. To this end, we introduce the Native-resolution diffusion Transformer (NiT), an architecture designed to explicitly model varying resolutions and aspect ratios within its denoising process. Free from the constraints of fixed formats, NiT learns intrinsic visual distributions from images spanning a broad range of resolutions and aspect ratios. Notably, a single NiT model simultaneously achieves the state-of-the-art performance on both *ImageNet*-$256 \times 256$ and $512 \times 512$ benchmarks. Surprisingly, akin to the robust zero-shot capabilities seen in advanced large language models, NiT, trained solely on ImageNet, demonstrates excellent zero-shot generalization performance. It successfully generates high-fidelity images at previously unseen high resolutions (*e.g.*, $1536 \times 1536$) and diverse aspect ratios (*e.g.*, $16:9, 3:1, 4:3$), as shown in Figure 1. These findings indicate the significant potential of native-resolution modeling as a bridge between visual generative modeling and advanced LLM methodologies.

## 1 Introduction

The emergence of Large Language Models (LLMs) [1, 6, 22, 26, 43, 49, 70, 74, 77, 78, 84] represents a transformative development in the AI-Generated Content (AIGC) area. Their success is largely attributed to two key characteristics: exceptional scalability and remarkable zero-shot generalization. Scalability is empirically validated by established scaling laws [1, 34], which demonstrate predictable performance gains with increased model size and dataset scale. Meanwhile, zero-shot generalization is evidenced by their capability to perform tasks for which they were not explicitly trained, such as seamlessly handling unseen questions of variable lengths, affording exceptional flexibility.

Concurrently, diffusion models [23, 30, 35, 36, 42, 47, 55, 57, 61, 66, 83, 85, 88, 89] have risen to prominence in visual generative modeling, lauded for their capacity to synthesize high-fidelity data. However, prevailing diffusion transformers [23, 47, 55] typically standardize images to fixed, often square, dimensions during training. This preprocessing step, while simplifying model architecture, inherently discards crucial native resolution and aspect ratio information. Such a practice curtails the models' ability to learn visual features across diverse scales and orientations [11, 46, 57, 80, 85], thereby limiting their intrinsic flexibility and generalization capabilities concerning input variability.

Large Language Models effectively process variable-length text by training directly on native data formats [1, 16, 22, 71, 72, 78, 84]. This inherent adaptability inspires a critical question for image synthesis: *Can diffusion models achieve similar flexibility, learning to generate images directly at their diverse, native resolutions and aspect ratios*? Conventional diffusion models face considerable challenges in generalizing across resolutions beyond their training regime. This limitation stems from three core difficulties: **1**) *Strong coupling between fixed receptive fields in convolutional architectures and learned feature scales* [18, 35, 36, 61]. These models internalize visual concepts at a resolution-specific scale, hindering effective feature extraction when resolution changes. **2**) *Fragility of positional encoding and spatial coordinate dependencies in transformer architectures* [25, 47, 55]. Hardcoded or learned positional encoding for a specific grid size, leads to distorted spatial reasoning and object coherence at novel resolutions. **3**) *Inefficient and unstable training dynamics from variable inputs.* Padding variable inputs [46, 81] causes waste and artifacts; while aspect ratio bucketing [11, 57, 82] increases training complexity. Both methods harm efficiency and the diffusion process's sensitivity to resolution-dependent image statistics. Addressing these interconnected challenges is crucial for developing truly native-resolution diffusion models.

In this work, we overcome these limitations by proposing a novel architecture for diffusion transformers that directly models native-resolution image data for generation. Drawing inspiration from the variable-sequence nature of Vision Transformers [16, 21], we reformulate image generative modeling within diffusion transformers as "native-resolution generation". And we present the Native-resolution diffusion Transformer (NiT), which demonstrates the capability to generate images across a wide spectrum of resolutions and aspect ratios. By exclusively training on images at their original resolutions, without resolution-modifying augmentations, our model inherently captures robust spatial relationships. NiT is developed based on the DiT [55] architecture and incorporates several key architectural innovations: **1**) *Dynamic Tokenization*. Images in native resolution are converted into variable-length token sequences and the corresponding height and width tuples. Without requiring input padding, it avoids substantial computational overhead. **2**) *Variable-Length Sequence Processing*. We use Flash Attention [15] to natively process heterogeneous, unpadded token sequences by

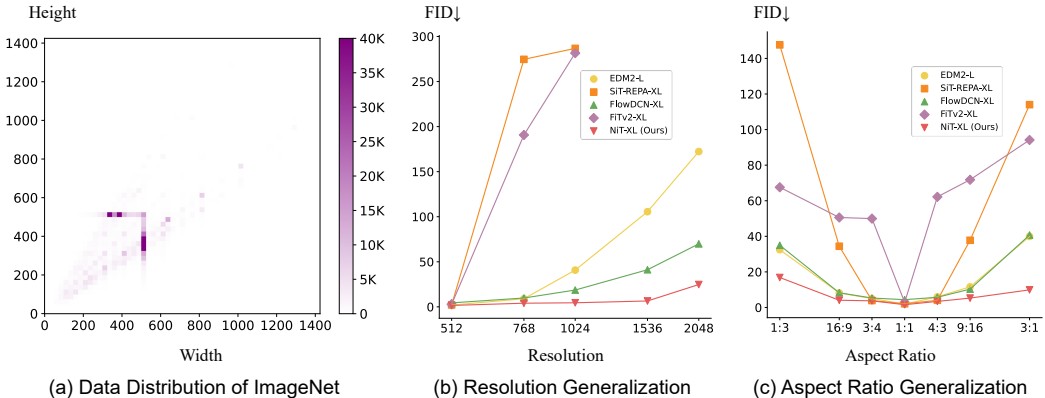

(a) Data Distribution of ImageNet   (b) Resolution Generalization   (c) Aspect Ratio Generalization

Figure 2: **NiT's Superior Generalization Beyond *ImageNet*'s Typical Resolution Distribution.** (a) *ImageNet* resolutions are mainly concentrated between 200 to 600 pixels (width/height), with sparse data beyond 800 pixels. Despite this, (b) shows our NiT model's superior generalization to unseen high resolutions (*e.g.*, 1024, 1536), evidenced by significantly lower FID scores. (c) further confirms NiT also exhibits the strongest generalization across various aspect ratios.

cumulative sequence lengths using the memory tiling strategy. **3**) *2D Structural Prior Injection*. We introduce the axial 2D Rotary Positional Embedding [68] (2D RoPE) to factorize the height and width impact and maximize the 2D structural prior by relative positional encoding.

Extensive experiments in class-guided image generation validate NiT as a significant advancement due to its native-resolution modeling. With **a single model**, NiT firstly attains state-of-the-art (SOTA) results on both $256 \times 256$ (2.03 FID, Fréchet inception distance) and $512 \times 512$ (**1.45** FID) benchmarks in class-guided *ImageNet* generation [17]. Impressively, NiT highlights its strong zero-shot generalization ability. For instance, as shown in Fig. 2, it achieves an FID of $4.52$ on unseen $1024 \times 1024$ resolution and an FID of $4.11$ on novel $9 : 16$ aspect ratio images (*i.e.*, $416 \times 768$), excelling in its flexibility and transferability to unfamiliar resolutions and respective ratios.

## 2   Related Work

### 2.1   Explorations of Variable-length Generalization

Large language models (LLMs) [1, 22, 49, 74, 77, 78, 84] are trained using text sequences with native, original length, which enables them to take flexible-length texts as input and generate arbitrary-length outputs. Seminal works, such as RoPE [68], NoPE [37], Alibi [58], and KERPLE [14], study the impact of positional encoding on the length generalization in language models. RoPE is then utilized in a wide range of LLMs as it unifies the relative position encoding with the absolute positional encoding. Beyond the valuable properties of RoPE, a series of works have been proposed to study the extreme context-length generalization of LLMs. NTK-RoPE [45], YaRN [56], LongRoPE [20], LM-Infinite [27], PoSE [91], and CLEX [9] explore the extreme length generalization of LLMs, enabling LLMs with the capability of a very long context length ($> 128K$) generation.

In the realm of computer vision, NaViT [16], DINO-v2 [54], and RADIO-v2.5 [28] have explored multi-resolution training to obtain a more robust vision representation. Recently, advanced vision language models (VLMs), including Qwen2-VL [4], Gemini1.5 [72], Intern-VL-2.5 [13], and Seed1.5-VL [70], have explored the native-resolution training in their vision encoders. However, the power of native-resolution training in visual content generation appears to be somewhat **locked**. In this work, we bridge the gap by exploring the native-resolution training in diffusion transformer models.

### 2.2   Explorations of Resolution Generalization in Visual Content Generation

Current visual generative models, whether autoregressive or diffusion-based, typically do not directly process visual content at its native resolution. Existing strategies to accommodate variable resolutions can be broadly categorized into three main approaches:

**Bucket Sampling** [11, 57, 82] facilitates dynamic resolution handling across different training batches by grouping samples into pre-defined "buckets". While the resolution and aspect ratio are fixed within each batch, they can vary between batches. The primary limitation is its reliance on these pre-defined buckets, restricting true flexibility to a discrete set of image dimensions.

**Padding and Masking** [46, 81] establishes a maximum sequence length, padding all image tokens to this length, and employing a mask to exclude padded regions from the loss calculation. This allows for dynamic resolution processing within a single batch. However, the approach often leads to significant computational and memory inefficiencies due to the processing of extensive padded areas, especially for images considerably smaller than the maximum resolution.

**Progressive Multi-Resolution** [10, 85] methods split the training process into several stages and progressively increase the image resolutions at each stage. While this method can effectively achieve high-resolution generation, it often exhibits suboptimal performance on smaller resolutions at earlier stages. Besides, it cannot generalize to higher resolutions beyond the last training stage.

The success of LLMs with variable-length inputs underscores a gap in visual content generation, where true native-resolution flexibility is still elusive. Existing methods to handle varied resolutions often introduce trade-offs in efficiency or generalization. To bridge this, we explore native-resolution training for diffusion transformer models, seeking a more fundamental solution to these persistent challenges. Our proposed methodology is detailed next.

## 3 Native-Resolution Diffusion Transformer

### 3.1 Preliminaries

**Diffusion Foundation.** Given the noise distribution $\epsilon \sim \mathcal{N}(0, \mathbf{I})$ and the data distribution $x \sim p_{\text{data}}(x)$, the time-dependent forward diffusion process is defined as: $x_t = \alpha_t x + \sigma_t \epsilon$, where $\alpha_t$ is a decreasing function of $t$ and $\sigma_t$ is an increasing function of $t$. There are different strategies to train a diffusion model, including DDPM [30, 52], score matching [32, 64, 65], EDM [35, 36], and flow matching [2, 3, 42, 44]. We adopt flow matching with linear path in NiT, which restricts the forward and reverse process on $t \in [0, 1]$ and set $\alpha_t = 1 - t, \sigma_t = t$, interpolating between the two distributions with velocity target $v = \epsilon - x$. The Logit-Normal time distribution $t = \frac{\sigma}{1+\sigma}$ from EDM is introduced, where $\ln(\sigma) \sim \mathcal{N}(P_{\text{mean}}, P_{\text{std}}^2)$ with manually selected coefficients $P_{\text{mean}}$ and $P_{\text{std}}$.

**Conventional Fixed-Resolution Modeling** The prevalent training strategy of diffusion models involves pre-setting a batch-wide image resolution, denoted as $H_{target} \times W_{target}$, often with $H_{target} = W_{target}$ for benchmarks like *ImageNet*-$256 \times 256$. Images $I_{orig}$ of native dimensions $H_{orig} \times W_{orig}$ are then subjected to resizing and cropping operations to conform. While simplifying model design, this fixed-resolution approach introduces three significant issues:

- *Spatial Structure and Semantic Degradation.* As discussed in previous works [33, 53, 67], resizing an image to a certain size via an interpolation function $f_{interp}(\cdot)$ with scaling factors $s_H, s_W$ can be detrimental. Upsampling (if $s_H > 1$ or $s_W > 1$) often introduces blurriness, diminishing sharpness. Conversely, downsampling (if $s_H < 1$ or $s_W < 1$) leads to an irrecoverable loss of high-frequency details. Subsequent cropping to $H_{target} \times W_{target}$, particularly when aspect ratios $\frac{W_{orig}}{H_{orig}} \neq \frac{W_{target}}{H_{target}}$, discards image regions, potentially leading to semantic incompleteness or contextual loss, an effect observed to influence generated samples in models, as revealed in SDXL [57].

- *Inhibited Resolution Generalization.* Models trained exclusively at a fixed resolution $(H_{target}, W_{target})$ exhibit poor generative performance at novel resolutions $(H_{infer}, W_{infer}) \neq (H_{target}, W_{target})$.[2] This limitation is particularly acute for Transformer-based diffusion models, like DiT [55] and SiT [47]. Their reliance on absolute positional embeddings lacks 2D structural modeling, which does not readily adapt to changes in the number or spatial arrangement of patches resulting from differing resolutions.

- *Inefficient Data Utilization and Computational Overhead.* Natural image datasets contain rich visual information encoded in their diverse native resolutions. Standardizing all inputs to $H_{target} \times W_{target}$ discards this inherent scale diversity [57]. Consequently, achieving high

---

[2]We provide a detailed comparison on this generalization ability in Table 2, and 3.

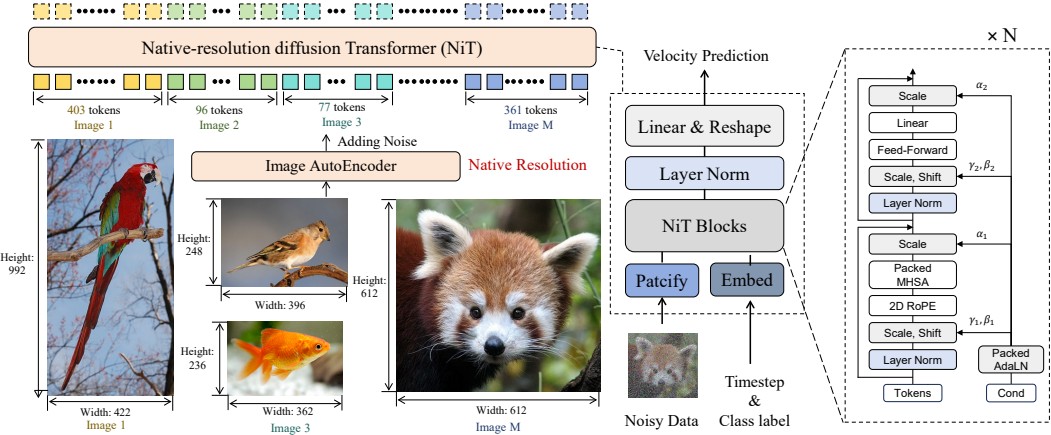

Figure 3: **Architecture Design of Native Resolution Diffusion Transformer (NiT).** NiT takes noisy latent representations, tokenizes them into variable-length sequences based on the original image resolution. Each NiT block utilizes Packed Multi-Head-Self-Attention (MHSA) with 2D RoPE and incorporates timestep and class conditioning via adaptive layer normalization.

performance across multiple resolutions necessitates training distinct model instances for each specific resolution. So, for different resolutions, current approaches incur a cumulative computational training cost.[3]

## 3.2 Native-Resolution Modeling

**Reformulation.** As illustrated in Fig. 3, given a sequence of images in arbitrary resolutions, we use the image autoencoder to compress the image sequence to a latent sequence $\boldsymbol{x} = \{x_1, x_2, x_3, \cdots, x_n\}$. Similar to LLMs [4], we pre-define a maximum sequence length $L$ and pack the latents $x_i \in \mathbb{R}^{c \times h_i \times w_i}$ together, where $c$, $h_i$ and $w_i$ are the dimension, height and width of the $i$-th latent, and $p$ is the patch size. Please note that in the packing algorithm, to maintain the maximum sequence length, image instance number $n$ is also dynamic in different iterations.

$$x_i \in \mathbb{R}^{c \times h_i \times w_i} \Rightarrow x_i \in \mathbb{R}^{(\frac{h_i \cdot w_i}{p^2}) \times (c \cdot p \cdot p)},$$
$$\boldsymbol{x} = \{x_1, x_2, x_3, \cdots, x_n\} \Rightarrow \boldsymbol{x} \in \mathbb{R}^{(\frac{1}{p^2} \Sigma_i h_i \cdot w_i) \times (c \cdot p \cdot p)}, \tag{1}$$

where the packing operation is to concatenate the variable-length latents to the maximum sequence length and improve computation efficiency.

After packing, we add noise and patchify the packed latent sequence $\boldsymbol{x}$. For each latent $x_i$, we sample each Gaussian noise in a variable-length independently, where the noise $\epsilon_i \in \mathbb{R}^{(\frac{h_i \cdot w_i}{p^2}) \times (c \cdot p \cdot p)}$ and time $t_i \in [0, 1]$ are added to the real data $x_i$:

$$x_i' = (1 - t_i) \cdot x_i + t_i \cdot \epsilon_i, \tag{2}$$

Then we use the patch embedding layers to project the noisy data to the visual tokens $\boldsymbol{z}$:

$$\mathbf{z}_0 = \texttt{PatchEmbed}(\boldsymbol{x}) \in \mathbb{R}^{(\frac{1}{p^2} \Sigma_i h_i \cdot w_i) \times d}, \tag{3}$$

where $d$ is the hidden size of the diffusion transformer. Thus, a sequence of native-resolution images is formulated into a packed sequence of visual tokens, and it satisfies: $(\frac{1}{p^2} \Sigma_i h_i \cdot w_i) \leqslant L$.

**Use axial 2D-RoPE to inject 2D structural priors.** The advantages of 2D Rotary Position Embedding (RoPE) for modeling inputs at their native resolutions are two-fold: **1**) 2D RoPE explicitly

---

[3]For example, despite SiT-REPA [88] achieving strong results on ImageNet-$256 \times 256$, a separate model, and thus roughly $2\times$ the training compute, are required for $512 \times 512$ resolution, highlighting an inefficient use of resources compared to models capable of handling variable resolutions.

[4]We adopt an efficient longest-pack-first histogram packing algorithm [39].

models 2D structural relationships within the image plane, offering better adaptability to various resolutions compared to learnable positional embeddings. **2)** The axial nature of 2D RoPE decouples height and width modeling, independently generating 1D rotational frequency components based on each token's patchfied height $h'_k$ and width $w'_k$. Let $\mathbf{z}_k$ be the tokens associated with height-width coordinates $(h'_k, w'_k)$. $d$ is the hidden size of query ($\mathbf{q}_k$) and key ($\mathbf{k}_k$) vectors derived from $\mathbf{z}_k$. The dimensionality of the rotary angle space is $d_s = d/2$. Base angular frequencies $\omega_j$ are defined as:

$$\omega_j = \theta^{-2j/d_s} \quad \text{for } j \in \{0, \ldots, d_s/2 - 1\}, \tag{4}$$

where $\theta$ is a hyperparameter.[5] The composite angle vector $\boldsymbol{\Phi}_{h'_k, w'_k} \in \mathbb{R}^{d_s}$ for the token at position $(h'_k, w'_k)$ is formed by concatenating angles derived from its height and width:

$$\boldsymbol{\Phi}_{h'_k, w'_k} = \texttt{Concat}\left(\{h'_k \cdot \omega_j\}_{j=0}^{d_s/2-1}, \{w'_k \cdot \omega_j\}_{j=0}^{d_s/2-1}\right). \tag{5}$$

In the self-attention mechanism, the query $\mathbf{q}_k \in \mathbb{R}^d$ and key $\mathbf{k}_k \in \mathbb{R}^d$ are transformed using these rotary embeddings. For any such vector $\mathbf{v} \in \mathbb{R}^d$ (representing either $\mathbf{q}_k$ or $\mathbf{k}_k$), its transformed version $\mathbf{v}'$ is obtained by rotating its feature pairs. Specifically, for each $l \in \{0, \ldots, d_s - 1\}$:

$$\begin{pmatrix} v'_{2l} \\ v'_{2l+1} \end{pmatrix} = \begin{pmatrix} \cos\left((\boldsymbol{\Phi}_{h'_k, w'_k})_l\right) & -\sin\left((\boldsymbol{\Phi}_{h'_k, w'_k})_l\right) \\ \sin\left((\boldsymbol{\Phi}_{h'_k, w'_k})_l\right) & \cos\left((\boldsymbol{\Phi}_{h'_k, w'_k})_l\right) \end{pmatrix} \begin{pmatrix} v_{2l} \\ v_{2l+1} \end{pmatrix}. \tag{6}$$

This axial 2D RoPE effectively injects 2D structural priors suitable for variable heights and widths, while inherently preserving RoPE's capacity to encode relative positions within the self-attention.

**Packed Full-Attention.** Processing packed sequences, which concatenate multiple variable-length visual token sequences from distinct data instances, necessitates restricting self-attention computations to operate only *within* the tokens of each original instance. While standard attention masking can enforce this, its application to the highly sparse structure of packed sequences incurs prohibitive computational overhead. Drawing inspiration from the efficient native-resolution processing in advanced Vision Language Models (VLMs) [4, 70, 72, 75], we employ FlashAttention-2 [15] to achieve this efficiently. Specifically, for a packed batch comprising $n$ data instances, where the $i$-th instance contributes $N_i = (h_i \cdot w_i)/p^2$ tokens (derived from its latent dimensions $h_i, w_i$ and patch size $p$), we define the individual token sequence lengths and their cumulative sums:

$$\texttt{CuSeqLens} = [0, N_1, N_1 + N_2, \ldots, \sum_{j=1}^{n-1} N_j, \sum_{j=1}^{n} N_j]. \tag{7}$$

This leverages FlashAttention-2's ability to handle packed inputs with variable sequence lengths (specified by $\texttt{CuSeqLens}$), thereby enabling efficient full-attention within each data instance without explicit masking or padding. A detailed algorithm is demonstrated in Algorithm 1

**Packed Adaptive Layer Normalization.** Conventional Adaptive Layer Normalization (AdaLN) methods are not directly suited for packed sequences due to the heterogeneity in sequence lengths and the need for instance-specific conditioning. To address this, we introduce Packed Adaptive Layer Normalization. For each $k$-th data instance within the packed sequence, its unique conditional embedding $\mathbf{c}_k \in \mathbb{R}^d$ (where $d$ is the token feature dimension) is utilized to modulate its corresponding visual tokens. Specifically, $\mathbf{c}_k$ is first projected to produce instance-specific scale ($\boldsymbol{\gamma}_k \in \mathbb{R}^d$) and shift ($\boldsymbol{\beta}_k \in \mathbb{R}^d$) parameters. These parameters are then broadcast across all $N_k = (h_k \cdot w_k)/p^2$ tokens originating from the $k$-th data instance. If $\hat{\mathbf{z}}_{k,j}$ is the $j$-th token of the $k$-th instance after standard Layer Normalization (applied to the entire packed sequence $\boldsymbol{z}$ to produce $\hat{\boldsymbol{z}}$), the AdaLN operation is:

$$\text{AdaLN}(\hat{\mathbf{z}}_{k,j}, \mathbf{c}_k) = \boldsymbol{\gamma}_k \odot \hat{\mathbf{z}}_{k,j} + \boldsymbol{\beta}_k, \tag{8}$$

where $\odot$ denotes element-wise multiplication. This ensures that adaptive normalization is applied consistently and specifically to each sub-sequence of tokens based on its original data instance, maintaining fidelity of conditioning within the computationally efficient packed representation.

**Conclusion.** NiT's architecture design of native-resolution generative modeling fundamentally enhances image synthesis. By preserving the complete spatial hierarchy and detail of original inputs, NiT intrinsically learns scale-independent visual distributions. This leads to superior fidelity and adaptability in zero-shot generalization across diverse resolutions and aspect ratios.

---

[5]Following the common practice [4, 46, 68, 70, 74, 85], we use $\theta = 10000$ to ensure distinct positional signals over typical sequence lengths.

Table 1: **Benchmarking class-conditional image generation on standard *ImageNet* $256 \times 256$ and $512 \times 512$ benchmarks.** Notably, **a single NiT model can compete on both two benchmarks**. "↓" or "↑" indicate lower or higher values are better. "# Res" and "# Token" respectively represent the resolution, total training token budget. "mFID" denotes the average Fréchet inception distance (FID) value of two benchmarks. "†": an independent model is required for each benchmark, leading to a cumulative computational cost, as reflected by the huge "# Token". All the results are reported with the utilization of classifier-free-guidance (CFG).

| Method | # Param | # Res | # Token | $256 \times 256$ | | | | | $512 \times 512$ | | | | | mFID↓ |
| | | | | FID↓ | sFID↓ | IS↑ | Prec.↑ | Rec.↑ | FID↓ | sFID↓ | IS↑ | Prec.↑ | Rec.↑ | |
| --- | --- | --- | --- | --- | --- | --- | --- | --- | --- | --- | --- | --- | --- | --- |
| *Auto-regressive Models for Specific Resolutions* | | | | | | | | | | | | | | |
| MaskGiT | - | 256 | - | 6.18 | - | 182.1 | 0.80 | 0.51 | - | - | - | - | - | - |
| LlamaGen-3B | 3B | 384 | $221B$ | 2.18 | 5.96 | 263.33 | 0.82 | 0.58 | - | - | - | - | - | - |
| ST-AR-XL | 775M | 256 | - | 2.37 | 6.05 | 270.59 | 0.82 | 0.58 | - | - | - | - | - | - |
| VAR-2B† | 2B | 256 | - | 1.73 | - | **350.2** | 0.82 | 0.60 | 2.63 | - | 303.2 | - | - | 2.18 |
| *Diffusion Models for Specific Resolutions* | | | | | | | | | | | | | | |
| DiT-XL/2† | 675M | 256&512 | $1428B$ | 2.27 | 4.60 | 278.24 | **0.83** | 0.57 | 3.04 | 5.02 | 240.82 | 0.84 | 0.54 | 2.66 |
| SiT-XL/2† | 675M | 256&512 | $1428B$ | 2.06 | 4.50 | 270.3 | 0.82 | 0.57 | 2.62 | 4.18 | 252.2 | 0.84 | 0.57 | 2.34 |
| FlowDCN† | 675M | 256&512 | $158B$ | 2.00 | **4.37** | 263.16 | 0.82 | 0.58 | 2.44 | 4.53 | 252.8 | 0.84 | 0.54 | 2.22 |
| FiTv2-XL† | 671M | 256&512 | $237B$ | 2.26 | 4.44 | 293.83 | 0.80 | 0.62 | 2.62 | 6.63 | **307.54** | 0.80 | 0.57 | 2.44 |
| SiT-REPA† | 675M | 256&512 | $525B$ | **1.42** | 4.70 | 305.7 | 0.80 | **0.65** | 2.08 | 4.19 | 274.6 | **0.83** | 0.58 | 1.75 |
| PixNerd† | 700M | 256&512 | - | 2.15 | 4.55 | 297.00 | 0.79 | 0.59 | 2.84 | 5.95 | 245.62 | 0.80 | 0.59 | 2.49 |
| EDM2-L | 777M | 512 | $472B$ | - | - | - | - | - | 1.88 | 4.27 | 258.21 | 0.81 | 0.58 | - |
| EDM2-XXL | 1.5B | 512 | $472B$ | - | - | - | - | - | 1.81 | - | - | - | - | - |
| *Generalist Diffusion Models for Arbitrary Resolution* | | | | | | | | | | | | | | |
| **NiT-XL** | 675M | Native | $131B$ | 2.16 | 6.34 | 253.44 | 0.79 | 0.62 | 1.57 | 4.13 | 260.69 | 0.81 | **0.63** | 1.86 |
| **NiT-XL** | 675M | Native | $197B$ | 2.03 | 6.28 | 265.26 | 0.80 | 0.62 | **1.45** | **4.04** | 272.77 | 0.81 | 0.62 | **1.74** |

# 4 Experiments

## 4.1 Setup

**Native-Resolution Generation Evaluation.**

To comprehensively evaluate the resolution generalization, we conduct a wide range of resolution spectra.

- We evaluate NiT on standard $256 \times 256$ and $512 \times 512$ benchmarks with **a single model**, differing from the previous implementations with two distinct models.
- For high-resolution generalization evaluation, experiments are conducted on four resolutions: $\{768 \times 768, 1024 \times 1024, 1536 \times 1536, 2048 \times 2048\}$.
- For aspect ratio generalization analysis, experiments are conducted on six aspect ratios: $\{1 : 3, 9 : 16, 3 : 4, 4 : 3, 16 : 9, 3 : 1\}$. The corresponding resolutions are: $\{320 \times 960, 416 \times 768, 480 \times 640, 640 \times 480, 768 \times 416, 960 \times 320\}$.

**Implementation Details.** We use DC-AE [12] with a $32\times$ down-sampling scale and 32 latent dimensions as our image encoder. Therefore, an image with the shape of $H \times W$ is encoded into a latent token vector $\mathbf{z} \in \mathbb{R}^{\frac{H}{32} \times \frac{W}{32} \times 32}$. For class-guided image generation experiments, the model architecture follows DiT [55], except for using a patch size of $1$. Our model is trained with native-resolution images, so the batch size is unsuitable in our setting. Therefore, similar to LLMs [22, 84], we use **token budget** (*i.e.*, the summation of total tokens in all training iterations) to represent the training compute. For class-guided image generation, we use $131,072^6$ tokens in one iteration. Unless otherwise stated, all results in Tabs. 1 to 3 are evaluated with the NiT model trained for $1000K$ steps (corresponds to $131B$ token budgets). We report FID [29], sFID [50], Inception Score (IS) [62], Precision and Recall [40] using ADM evaluation suite [18]. Text-to-image generation experiments are detailed in Appendix A.

## 4.2 State-of-the-Art Class-Guided Image Generation

**Standard Benchmarks**

We first demonstrate the effectiveness of NiT on standard *ImageNet* $256 \times 256$ and $512 \times 512$ benchmarks. We compare NiT with state-of-the-art autoregressive models: MaskGit [8], Llam-

---

[6]It holds: $131,072 = 256 \times 512 = 1024 \times 128$. For $256 \times 256$ resolution (256 tokens each in DiT [55]), this corresponds to a larger batch of 512 images. Conversely, for higher-resolution $512 \times 512$ images (1024 tokens each), it corresponds to a smaller batch of 128 images is used to maintain the same total token count.

Table 2: **Benchmarking resolution generalization capabilities on *ImageNet*.** As FiTv2 and SiT-REPA have failed to generalize to $1024 \times 1024$ resolution, we use "✗" to represent their inabilities to generalize to higher resolutions.

| Method | $768 \times 768$ | | | $1024 \times 1024$ | | | $1536 \times 1536$ | | | $2048 \times 2048$ | | |
|---|---|---|---|---|---|---|---|---|---|---|---|---|
| | FID↓ | sFID↓ | IS↑ | FID↓ | sFID↓ | IS↑ | FID↓ | sFID↓ | IS↑ | FID↓ | sFID↓ | IS↑ |
| EDM2-L | 9.02 | 18.57 | 248.15 | 40.74 | 47.29 | 119.41 | 105.57 | 69.31 | 40.05 | 172.30 | 89.18 | 16.82 |
| FlowDCN | 9.817 | 24.52 | 202.86 | 18.64 | 42.36 | 206.66 | 41.170 | 61.75 | 150.22 | 69.88 | 68.15 | 81.33 |
| FiTv2-XL | 190.69 | 143.78 | 8.56 | 281.55 | 209.20 | 4.55 | ✗ | ✗ | ✗ | ✗ | ✗ | ✗ |
| SiT-REPA | 274.63 | 215.25 | 3.58 | 286.79 | 235.07 | 2.643 | ✗ | ✗ | ✗ | ✗ | ✗ | ✗ |
| **NiT-XL** | **4.05** | **8.77** | **262.31** | **4.52** | **7.99** | **286.87** | **6.51** | **9.97** | **230.10** | **24.76** | **18.01** | **131.36** |

Table 3: **Benchmarking aspect ratio generalization capabilities on *ImageNet*.**"†": SiT-REPA is evaluated with $160 \times 480, 216 \times 384, 240 \times 320, 320 \times 240, 384 \times 216$ and $480 \times 160$, because only the model trained on 256-resolution image data is open-sourced. For other models, the exact resolutions are: $320 \times 960, 416 \times 768, 480 \times 640, 640 \times 480, 768 \times 416$ and $960 \times 320$.

| Method | $1:3$ | | $9:16$ | | $3:4$ | | $4:3$ | | $16:9$ | | $3:1$ | |
|---|---|---|---|---|---|---|---|---|---|---|---|---|
| | FID↓ | IS↑ | FID↓ | IS↑ | FID↓ | IS↑ | FID↓ | IS↑ | FID↓ | IS↑ | FID↓ | IS↑ |
| EDM2-L | 32.48 | 68.45 | 8.19 | 170.06 | 5.00 | 183.78 | 5.97 | 170.06 | 11.58 | 144.65 | 39.94 | 59.69 |
| FlowDCN | 34.97 | 73.30 | 8.25 | 178.16 | 5.311 | 197.01 | 5.74 | 185.67 | 10.31 | 154.92 | 40.721 | 64.11 |
| FiTv2-XL | 67.57 | 37.95 | 50.58 | 60.52 | 49.96 | 59.40 | 62.18 | 46.06 | 71.79 | 42.89 | 94.15 | 35.74 |
| SiT-REPA | 147.61 | 14.44 | 34.42 | 94.13 | 3.87 | 242.61 | 4.03 | 242.42 | 37.77 | 88.46 | 114.01 | 18.56 |
| **NiT-XL** | **16.85** | **189.18** | **4.11** | **254.71** | **3.72** | **284.94** | **3.41** | **259.06** | **5.27** | **218.78** | **9.90** | **255.05** |

aGen [69], VAR [76], and ST-AR [90], as well as state-of-the-art diffusion models: DiT [48], SiT [47], FlowDCN [80], PixelNerd [79], FiTv2 [81], SiT-REPA [88], and EDM2 [36]. All these are resolution-expert methods, independently training two models for the two benchmarks.

**Performance Analysis.** As demonstrated in Tab. 1, NiT-XL achieves **the best FID** $1.45$ **on the** $512 \times 512$ **benchmark**, outperforming the previous SOTA EDM2-XXL with half of the model size. On the $256 \times 256$ benchmark, our model surpasses the SiT-XL, DiT-XL, and FiTv2-XL models on FID with the same model size as well as outperforms the LlamaGen-3B model with much smaller parameters. Compared with all the baseline models, our model demonstrates **significant training efficiency**, as it avoids the cumulative computes to train two distinct models. To the best of our knowledge, this is the first time **a single model** can compete on these two benchmarks simultaneously. For *mFID* metric, NiT-XL largely outperforms DiT-XL and SiT-XL with $9.17\%$ of the training costs. And it can surpass SiT-REPA, with only $37.52\%$ of the token budget.

Beyond standard benchmarks on $256 \times 256$ and $512$ resolutions, we comprehensively conduct high-resolution generalization in Tab. 2 and aspect ratio generalization evaluations in Tab. 3. We compare our method with 4 widely-recognized baselines: 1) EDM2 [36], SOTA CNN-based diffusion model; 2) FlowDCN [80], CNN-based diffusion model designed for multi-resolution image generation;

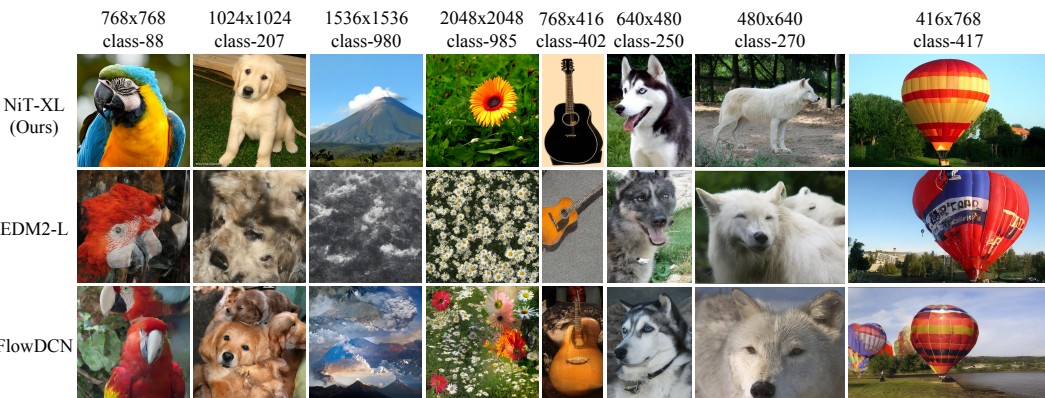

Figure 4: **Qualitative Comparison of Resolution and Aspect Ratio Generalization.** We provide the visualization of NiT, EDM2 and FlowDCN, because FiTv2 and SiT-REPA demonstrate inferior generalization capability revealed by quantitative results.

Table 4: **Ablation study on resolution generalization.** We explore different data mixtures to realize why NiT can generalize to unseen resolutions. All these models are trained for $100K$ Steps.

| Data Mixture | $256 \times 256$ | | | $512 \times 512$ | | | $768 \times 768$ | | |
|---|---|---|---|---|---|---|---|---|---|
| | FID↓ | sFID↓ | IS↑ | FID↓ | sFID↓ | IS↑ | FID↓ | sFID↓ | IS↑ |
| (a) Below 256 | 21.08 | 13.47 | 68.32 | 180.27 | 72.65 | 10.16 | 241.09 | 129.49 | 5.46 |
| (b) Below 512 | 36.10 | 14.95 | 35.77 | 28.06 | 14.23 | 58.22 | 195.99 | 83.87 | 10.00 |
| (c) Native Resolution | 31.95 | 16.69 | 41.31 | 24.52 | 12.84 | 65.89 | 25.68 | 13.89 | 70.56 |
| (d) Native Resolution + 256 + 512 | 24.54 | 8.39 | 53.30 | 17.62 | 5.77 | 81.64 | 21.03 | 11.75 | 86.25 |
| (e) 256 + 512 | 28.22 | 8.44 | 46.71 | 20.21 | 5.98 | 72.17 | 50.28 | 91.45 | 43.86 |
| (f) 512 | 127.47 | 53.72 | 10.60 | 24.64 | 6.18 | 59.14 | 86.36 | 124.91 | 19.43 |

Table 5: **Ablation study on resolution and aspect ratio generalization.** We further evaluate the data mixture on different aspect ratios. All these models are trained for $500K$ Steps.

| Data | $1024 \times 1024$ | | $1536 \times 1536$ | | $2048 \times 2048$ | | $320 \times 960$ | | $416 \times 768$ | | $480 \times 640$ | |
|---|---|---|---|---|---|---|---|---|---|---|---|---|
| | FID↓ | IS↑ | FID↓ | IS↑ | FID↓ | IS↑ | FID↓ | IS↑ | FID↓ | IS↑ | FID↓ | IS↑ |
| (c) | 13.15 | 159.52 | 18.11 | 132.53 | 41.95 | 71.19 | 40.14 | 53.41 | 13.45 | 137.31 | 10.63 | 160.91 |
| (d) | 9.73 | 180.29 | 14.29 | 152.37 | 37.79 | 78.66 | 43.86 | 50.25 | 12.14 | 131.96 | 8.55 | 169.24 |

3) FiTv2 [81], diffusion transformer model designed for multi-resolution image generation; 4) SiT-REPA [88], current SOTA diffusion transformer model.

**Generalization Analysis** As demonstrated in Tab. 2, NiT-XL significantly surpasses all the baselines on resolution generalization. Remarkably, NiT-XL achieves FID 4.07, 4.52, and 6.51 on 768, $1024 \times 1024$, and $1536 \times 1536$, respectively, demonstrating almost no performance degradation when scaling to unseen higher resolutions. EDM2-L and FlowDCN can generalize to $768 \times 768$ resolution, but they fail to generalize to higher resolutions beyond $1024 \times 1024$ resolution. FiTv2 and SiT-REPA demonstrate very inferior resolution generalization capability beyond their training resolutions. As shown in Tab. 3, NiT-XL can also generalize to arbitrary aspect ratios, greatly outperforming all the baselines. Although EDM2-L, SiT-REPA and FlowDCN can generalize to $4 : 3$ and $3 : 4$ aspect ratios, they fail to generalize more extreme aspect ratios, like $9 : 16$ and $1 : 3$. NiT-XL can generalize up to $9 : 16$ and $16 : 9$ aspect ratios with negligible performance loss, and perform best on $1 : 3$ and $3 : 1$ aspect ratios. These results indicate **NiT is initially equipped with the resolution-free generation capability**, bridging the gap between vision generation and the sequence-free generation in LLMs [1, 49, 74, 84].

The qualitative comparison in Fig. 4 is consistent with the aforementioned quantitative results. NiT demonstrates superior generalization quality to EDM2-L and FlowDCN, producing reasonable generated samples. When beyond 768 resolution, EDM2-L in particular generates images dominated by non-informative textures, while FlowDCN-XL tends to replicate objects multiple times in a single image. Regarding aspect ratio generalization, models like DM2-L and FlowDCN-XL exhibit a distinct cropping-induced bias. This strongly suggests the models have internalized a truncation bias due to their training predominantly on square or tightly cropped image samples. Consequently, they tend to generate unnaturally framed outputs with truncated object boundaries, especially when encountering extreme aspect ratios. This aligns with findings from SDXL [57], revealing that image cropping during data preparation can propagate biases into generated samples, leading to adverse effects, particularly with extreme aspect ratios. More visualizations are provided in Appendix D

## 4.3 Ablation Study

**Set up.** We find the surprising generalization ability of NiT in unseen resolutions and aspect ratios. In this part, we explore what enables the impressive generalization ability. We conduct 6 groups for ablation: (a): we keep the aspect ratio of image data and resize its height and width smaller than 256;(b): we keep the aspect ratio of image data and resize its height and width smaller than 512; (c): we only use native-resolution image data for training; (d): besides native-resolution, we add $256 \times 256$-resolution, and $512 \times 512$-resolution image data for training; (e): without native-resolution images, we only use $256 \times 256$ and $512 \times 512$ images; (f): we only use fixed $512$ resolution images. All the experiments are conducted using a NiT-B model with $131M$ parameters, and all the results are evaluated with the usage of CFG (CFG scale is set as $1.5$).

Table 6: **Ablation study on data and positional embeddding.** [†]: corresponds to SiT [47]. [‡]: corresponds to NiT. All the models are trained for $100K$ Steps.

| Data | Positional Embedding | $256 \times 256$ | | | $512 \times 512$ | | | $768 \times 768$ | | |
|---|---|---|---|---|---|---|---|---|---|---|
| | | **FID↓** | **sFID↓** | **IS↑** | **FID↓** | **sFID↓** | **IS↑** | **FID↓** | **sFID↓** | **IS↑** |
| (d)[†] | Sin-Cos PE | 35.66 | 8.76 | 35.65 | 29.41 | 6.60 | 51.26 | 31.06 | 12.69 | 59.15 |
| (d) | RoPE | 24.54 | 8.39 | 53.30 | 17.62 | 5.77 | 81.64 | 21.03 | 11.75 | 86.25 |
| (f) | Sin-Cos PE | 125.40 | 57.82 | 9.26 | 29.40 | 6.61 | 53.73 | 124.51 | 126.33 | 12.01 |
| (f)[‡] | RoPE | 127.47 | 53.72 | 10.60 | 24.64 | 6.18 | 59.14 | 86.36 | 124.91 | 19.43 |

As demonstrated in Tab. 4, (c) consistently beat (a) and (b) on resolution generalization. When training images are capped at 256 pixels on the long side (a), the model excels at 256 but degrades sharply at $512 \times 512$ and $768 \times 768$. Raising the cap to 512 pixels (b) postpones, but does not eliminate the drop-off: once the test resolution exceeds the highest resolution encountered in training, FID climbs rapidly. By contrast, training at native image resolution with no upper bound (c) yields consistently strong performance across the full range of resolutions we evaluated. Further, (d) and (e) consistently beat (c) on $256 \times 256$ and $512 \times 512$ benchmarks. This is because the model is not optimized for these two resolutions in (c), thus demonstrating inferior performance. The performance on $256 \times 256$ and $512 \times 512$ of (d) and (e) is comparable; however, (e) extremely lags (d) on $768 \times 768$ resolution generation. We think that the training of (e) is solely on two resolutions, significantly restricting the generalization capabilities of models. Meanwhile, in Tab. 5, as we scale up training steps to 500k, we further compare the performance of (c) and (d) on resolution generalization and aspect ratio generalization. We find that (d) demonstrates stronger generalization capability than (c).

**Positional Embedding Comparison.** Tab. 6 demonstrates the effects of data and positional embedding on resolution generalization. When using native-resolution data (f), both using Sin-Cos PE and using RoPE can demonstrate strong generalization capability across multiple resolutions. For both fixed-resolution settings and native-resolution settings, RoPE can improve model performance across multiple resolutions, indicating its effectiveness on 2D structural prior injection.

**Insights.** The ablation study reveals that NiT's strong generalization to unseen resolutions and aspect ratios is primarily enabled by training in native-resolution to learn a resolution- and aspect-ratio invariant visual distribution. While adding fixed resolutions like $256 \times 256$ and $512 \times 512$ improves performance on those specific sizes and aspect ratios of $1:1, 768 \times 768$, omitting native-resolution data severely hinders generalization to other aspect ratios. Therefore, a combination of varied resolution and aspect ratios, with native resolution playing a key role, is essential for robust generalization, rather than just training on a limited set of fixed scales.

**Efficiency Analysis.** We compare the training and inference efficiency on the ImageNet-256 benchmark using a single NVIDIA A100 GPU, revealing that NiT demonstrates better training and inference efficiency compared to DiT. Analysis is conducted with the NiT-B and DiT-B model, both with $131M$ parameters. We set the token number in one iteration as 65536. Specifically, NiT achieves a faster training speed of 1.28 iterations/second (iter/s) compared to DiT's 1.08 iter/s. Furthermore, NiT exhibits lower inference latency at 0.246 seconds, while DiT has a latency of 0.322 seconds, highlighting NiT's greater computational efficiency.

## 5    Conclusion & Limitation

In conclusion, this work introduces the native-resolution image synthesis paradigm for visual content generation. We reformulate the resolution modeling as "native-resolution generation" and propose Native-resolution diffusion Transformer (NiT). To the best of our knowledge, **NiT is the first model that can achieve dual SOTA performance on** $256 \times 256$ **and** $512 \times 512$ **benchmarks in** *ImageNet* **generation with a single model.** We demonstrate NiT's robust generalization capabilities to unseen image resolutions and aspect ratios, significantly outperforming previous methods. While NiT shows strong performance, its generalization ability on extremely high-resolution and aspect ratios is still not satisfactory. Future research could explore optimal strategies for balancing diverse training resolutions for improved efficiency and further investigate the model's generalization limits across a wider spectrum of data domains (*e.g.*, video generation) and highly disparate aspect ratios. Additionally, the computational resources required for comprehensive multi-scale training remain a practical consideration for broader application.

## Acknowledgements

This work was supported by the JC STEM Lab of AI for Science and Engineering, funded by The Hong Kong Jockey Club Charities Trust, the Research Grants Council of Hong Kong (Project No. CUHK14213224).

This work is partially supported by the National Natural Science Foundation of China (Grant No. 62306261), and the SHIAE Grant (No. 8115074). This study was supported in part by the Centre for Perceptual and Interactive Intelligence, a CUHK-led InnoCentre under the InnoHK initiative of the Innovation and Technology Commission of the Hong Kong Special Administrative Region Government. This work is also partially supported by Hong Kong RGC Strategic Topics Grant STG1/E-403/24-N.

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

# Appendices

## A   Text-to-Image Generation

### A.1   Streamlined NiT Architecture for Text-to-Image Generation

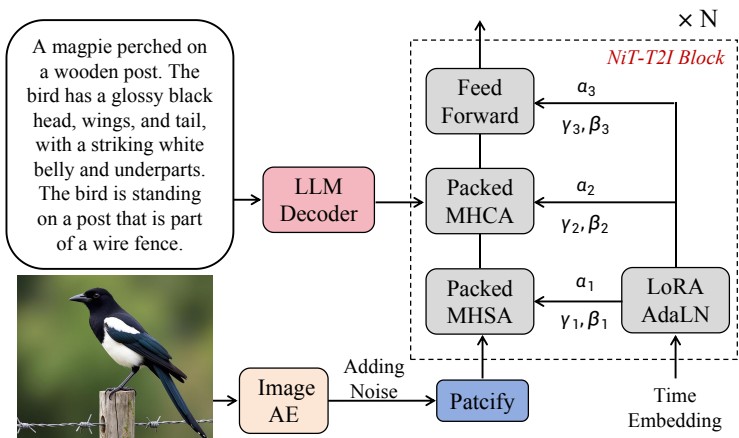

Figure 5: Illustration of NiT blocks used for text-to-image generation.

We introduce a streamlined architecture to incorporate textual information. As in Fig. 5, we insert a cross-attention block between the self-attention block and the feed-forward network. As the adaptive LayerNorm (AdaLN) module only conditions on the time embedding, we thus reduce its parameters through a LoRA [31] design. Given transformer hidden size $d$, the AdaLN layer predicts a tuple of all scale and shift parameters $S = [\beta_1, \beta_2, \beta_3, \gamma_1, \gamma_2, \gamma_3, \alpha_1, \alpha_2, \alpha_3]$, where $\beta, \gamma, \alpha$ represents the shift and scale parameters for a block, and the subscript $1, 2, 3$ denote self-attention, cross-attention, and feed-forward respectively. Based on the embedding for time step $\mathbf{t} \in \mathbb{R}^d$, $S^l$ for the $l$-th NiT-T2I block is computed as:

$$S^l = [\beta_1^l, \beta_2^l, \beta_3^l, \gamma_1^l, \gamma_2^l, \gamma_3^l, \alpha_1^l, \alpha_2^l, \alpha_3^l] = W_2^l W_1^l \mathbf{t} \in \mathbb{R}^{9 \times d}, \tag{9}$$

where $W_2^l \in \mathbb{R}^{(9 \times d) \times r}, W_1^l \in \mathbb{R}^{r \times d}$, and the bias parameters are omitted for simplicity. We can adjust the LoRA rank $r$ to align the block parameters with the NiT-C2I blocks.

### A.2   Advanced Text-to-Image Generation

**Implementation details.**   For text-to-image generation, and adopt Gemma3-1B-it [73] as our text encoder. We use LoRA rank $r = 192$ in AdaLN, which leads to a total $673M$ model parameters, matching the model parameters of the XL model in the C2I (class-to-image) setting. We use the REPA [88] strategy in the training, where a RADIO-v2.5-H [28] model serves as the visual encoder. We conduct text-to-image generation experiments on the SAM [38] dataset with captions generated by MiniCPM-V [86]. We use a token number in each iteration as $786, 416$ and train the model for $400K$ steps and evaluate image quality using COCO-val-2014 [41] benchmark.

**Compared baselines and Results.**   We report the zero-shot text-to-image performance on *COCO-2014* benchmark in Tab. 7, competing with DALL·E [60], CogView2 [19], Parti [87], Make-A-Scene [24], Muse [7], GLIDE [51], DALL·E 2 [59], LDM [61], eDiff-I [5], SD-v1.5 [61], and SDXL-Turbo [63]. NiT-T2I demonstrates competitive performance with the baseline models. Notably, NiT-T2I achieves the best CLIP score of $0.345$, and it surpasses SD-v1.5 and SDXL-Turbo on both FID and CLIP scores with a much smaller model size and training costs.

Table 7: **Text-to-image generation of NiT**. We compare our models by the zero-shot generation task on the *COCO*-2014 [41] dataset.

| Method | # Param. | Res. | FID (↓) | CLIP (↑) |
|---|---|---|---|---|
| DALL·E | 12B | 256 | 27.5 | - |
| CogView2 | 6B | 256 | 24.0 | - |
| Parti-750M | 750M | 256 | 10.71 | - |
| Parti-3B | 3B | 256 | 8.10 | - |
| Parti-20B | 20B | 256 | **7.23** | - |
| Make-A-Scene | - | 256 | 11.84 | - |
| Muse | 3B | 256 | 7.88 | 0.32 |
| GLIDE | 5B | 256 | 12.24 | - |
| DALL·E 2 | 5.5B | 256 | 10.39 | - |
| LDM | 1.45B | 256 | 12.63 | - |
| Imagen | 3B | 256 | 7.27 | - |
| eDiff-I | 9B | 256 | 6.95 | - |
| SD-v1.5 | 0.9B | 512 | 9.78 | 0.318 |
| SDXL-Turbo | 3B | 1024 | 23.19 | 0.334 |
| **NiT-T2I** | 673M | 1024 | 9.18 | **0.345** |

# B  Detailed Quantitative Results

In this section, we report all the metrics of generalization experiments in Tab. 8. In addition, we provide the CFG (classifier-free-guidance) hyperparameters of NiT-XL.

Table 8: **Detailed Quantitative Results of NiT-XL.** We further provide the CFG scale and interval for each experiment and report all the metric values for generalization experiments.

| Resolution | CFG-scale | CFG-interval | FID↓ | sFID↓ | IS↑ | Prec.↑ | Rec.↑ |
|---|---|---|---|---|---|---|---|
| $256 \times 256$ | 2.25 | [0.0, 0.7] | 2.16 | 6.34 | 253.44 | 0.79 | 0.62 |
| $512 \times 512$ | 2.05 | [0.0, 0.7] | 1.57 | 4.13 | 260.69 | 0.81 | 0.63 |
| $768 \times 768$ | 3.0 | [0.0, 0.7] | 4.05 | 8.77 | 262.31 | 0.83 | 0.52 |
| $1024 \times 1024$ | 3.0 | [0.0, 0.8] | 4.52 | 7.99 | 286.87 | 0.82 | 0.50 |
| $1536 \times 1536$ | 1.5 | [0.0, 1.0] | 6.51 | 9.97 | 230.10 | 0.83 | 0.42 |
| $2048 \times 2048$ | 1.5 | [0.0, 1.0] | 24.76 | 18.02 | 131.36 | 0.67 | 0.46 |
| $320 \times 960$ | 4.0 | [0.0, 0.9] | 16.85 | 17.79 | 189.18 | 0.71 | 0.38 |
| $416 \times 768$ | 2.75 | [0.0, 0.7] | 4.11 | 10.30 | 254.71 | 0.83 | 0.55 |
| $480 \times 640$ | 2.75 | [0.0, 0.7] | 3.72 | 8.23 | 284.94 | 0.83 | 0.54 |
| $640 \times 480$ | 2.5 | [0.0, 0.7] | 3.41 | 8.07 | 259.06 | 0.83 | 0.56 |
| $768 \times 416$ | 2.85 | [0.0, 0.7] | 5.27 | 9.92 | 218.78 | 0.80 | 0.55 |
| $960 \times 320$ | 4.5 | [0.0, 0.9] | 9.90 | 25.78 | 255.95 | 0.74 | 0.40 |

# C  Detailed Implementation of NiT

This section demonstrates the detailed implementation of Packed Full-Attention and Packed Adaptive Layer Normalization (AdaLN) in an NiT block. Different from traditional attention implementation for batched data, we use FlashAttention2 [15] for packed data, leading to enhanced efficiency, as in Algorithm 1. Besides, we use a broadcast mechanism on conditional vector **c** for Packed-AdaLN, detailed in Algorithm 2.

# D  Qualitative Results of NiT

## D.1  Generalization Comparison with Baseline Models

We provide the qualitative results of NiT-XL, EDM2-L [36], and FlowDCN-XL [80] on resolution generalization and aspect ratio generalization. The qualitative results of FiTv2-XL [81] and

**Algorithm 1** Packed Full-Attention with FlashAttention for flexible-length sequence processing.

```
1
2  import torch
3  import torch.nn as nn
4  from flash_attn import flash_attn_varlen_func
5
6  def rotate_half(x):
7      x = rearrange(x, '... (d r) -> ... d r', r = 2)
8      x1, x2 = x.unbind(dim = -1)
9      x = torch.stack((-x2, x1), dim = -1)
10     return rearrange(x, '... d r -> ... (d r)')
11
12 class Attention(nn.Module):
13     def __init__(self,
14         dim: int,
15         num_heads: int = 8,
16         qkv_bias: bool = False,
17         qk_norm: bool = False,
18         attn_drop: float = 0.,
19         proj_drop: float = 0.,
20         norm_layer: nn.Module = nn.LayerNorm,
21     ) -> None:
22         super().__init__()
23         assert dim % num_heads == 0, "dim should be divisible by num_heads"
24         self.num_heads = num_heads
25         self.head_dim = dim // num_heads
26         self.scale = self.head_dim ** -0.5
27         self.qkv = nn.Linear(dim, dim * 3, bias=qkv_bias)
28         self.q_norm = norm_layer(self.head_dim) if qk_norm else nn.Identity()
29         self.k_norm = norm_layer(self.head_dim) if qk_norm else nn.Identity()
30         self.attn_drop = nn.Dropout(attn_drop)
31         self.proj = nn.Linear(dim, dim)
32         self.proj_drop = nn.Dropout(proj_drop)
33
34     def forward(self,
35         x: torch.Tensor,
36         cu_seqlens: torch.Tensor,
37         freqs_cos: torch.Tensor,
38         freqs_sin: torch.Tensor
39     ) -> torch.Tensor:
40         # x: packed sequence with shape [N, D]
41         # cu_seqlens: [0, h_1*w1, h_1*w_1+h_2*w_2, ...], the cumulated sequence length
42         # freqs_cos, freqs_sin: 2D-RoPE frequences
43         N, C = x.shape
44         qkv = self.qkv(x).reshape(
45             N, 3, self.num_heads, self.head_dim
46         ).permute(1, 0, 2, 3)
47         ori_dtype = qkv.dtype
48         q, k, v = qkv.unbind(0)
49         q, k = self.q_norm(q), self.k_norm(k)
50
51         # Use axial 2D-RoPE to inject 2D structural priors
52         q = q * freqs_cos + rotate_half(q) * freqs_sin
53         k = k * freqs_cos + rotate_half(k) * freqs_sin
54         q, k = q.to(ori_dtype), k.to(ori_dtype)
55
56         max_seqlen = (cu_seqlens[1:] - cu_seqlens[:-1]).max().item()
57
58         # apply flash-attn for efficient implementation
59         x = flash_attn_varlen_func(
60             q, k, v, cu_seqlens, cu_seqlens, max_seqlen, max_seqlen
61         ).reshape(N, -1)
62
63         x = self.proj(x)
64         x = self.proj_drop(x)
65         return x
```

**Algorithm 2** Packed Adaptive Layer Normalization and NiT block.

```python
1
2  import torch
3  import torch.nn as nn
4  from timm.models.vision_transformer import Mlp
5
6
7  def modulate(x, shift, scale):
8      return x * (1 + scale) + shift
9
10 class NiTBlock(nn.Module):
11     """
12     A NiT block with adaptive layer norm zero (adaLN-Zero) conditioning.
13     """
14     def __init__(self, hidden_size, num_heads, mlp_ratio=4.0, **block_kwargs):
15         super().__init__()
16         self.norm1 = nn.LayerNorm(hidden_size, elementwise_affine=False, eps=1e-6)
17         self.attn = Attention(
18             hidden_size, num_heads=num_heads, qkv_bias=True,
19             qk_norm=block_kwargs["qk_norm"]
20         )
21         self.norm2 = nn.LayerNorm(hidden_size, elementwise_affine=False, eps=1e-6)
22         mlp_hidden_dim = int(hidden_size * mlp_ratio)
23         approx_gelu = lambda: nn.GELU(approximate="tanh")
24         self.mlp = Mlp(
25             in_features=hidden_size, hidden_features=mlp_hidden_dim,
26             act_layer=approx_gelu, drop=0
27         )
28         self.adaLN_modulation = nn.Sequential(
29             nn.SiLU(),
30             nn.Linear(hidden_size, 6 * hidden_size, bias=True)
31         )
32
33     def forward(self, x, c, hw_list, freqs_cos, freqs_sin):
34         # x: packed sequence with shape [N, D]
35         # c: conditional vector with shape [n, D], (n represetns number of instances)
36         # hw_list: [[h1_, w_1], [h2, w_2], ..., [h_n, w_n]]
37         # freqs_cos, freqs_sin: 2D-RoPE frequences
38
39         seqlens = hw_list[:, 0] * hw_list[:, 1]
40         cu_seqlens = torch.cat([
41             torch.tensor([0], device=hw_list.device, dtype=torch.int),
42             torch.cumsum(seqlens, dim=0, dtype=torch.int)
43         ])
44         # (n, D) -> (N, D) for Packed-AdaLN
45         c = torch.cat([c[i].unsqueeze(0).repeat(seqlens[i], 1) for i in range(B)], dim=0)
46
47         # predict all the shift-and-scale parameters with _acked-AdaLN
48         (
49             shift_msa, scale_msa, gate_msa, shift_mlp, scale_mlp, gate_mlp
50         ) = self.adaLN_modulation(c).chunk(6, dim=-1)
51
52         x = x + gate_msa * self.attn(
53             modulate(self.norm1(x), shift_msa, scale_msa),
54             cu_seqlens, freqs_cos, freqs_sin
55         )
56         x = x + gate_mlp * self.mlp(modulate(self.norm2(x), shift_mlp, scale_mlp))
57
58         return x
```

REPA-XL [88] are not provided because these two methods demonstrate very weak generalization capabilities. The resolution generalization visualizations are shown in Figs. 6 to 9, while the aspect ratio generalization visualizations are demonstrated in Figs. 10 to 15.

Based on these visualization results, we find that NiT-XL achieves the best qualitative performance on both resolution generalization and aspect ratio generalization, **consistent with its best FIDs and IS scores** in the manuscript. We then provide more analysis on the qualitative results.

**Analysis of resolution generalization visualization.** As demonstrated in Figs. 6 to 9, NiT-XL demonstrates almost no quality degradation from $768 \times 768$ to $1536 \times 1536$ resolution. It can also generate reasonable content on $2048 \times 2048$ resolution. However, EDM2-L and FlowDCN-XL demonstrate inferior visual quality in resolution generalization. Although EDM2-L and FlowDCN-XL can generate some plausible samples on $768 \times 768$ resolution, they fail to generalize to higher resolutions ($1024 \times 1024$ to $2048 \times 2048$). The key limitations are:

1. *Lack of Semantic Coherence.* Both EDM2-L and FlowDCN-XL frequently fail to generate identifiable, realistic instances of the target classes, especially for high resolution (see Fig. 9).

2. *Repetitive Textures.* EDM2-L in particular generates images dominated by repetitive, non-informative textures, lacking structural variation or clear object boundaries.

3. *Object Duplication and Spatial Disruption.* FlowDCN-XL tends to replicate objects multiple times in a single image, resulting in cluttered and spatially implausible compositions.

4. *Color and Lighting Artifacts.* EDM2-L often outputs grayscale or dull images, while FlowDCN-XL introduces unnatural color schemes and poor lighting consistency.

These limitations reveal that neither model effectively integrates objects into realistic backgrounds, with EDM2-L omitting context and FlowDCN-XL producing jumbled scenes. There is **an evident reliance on local textures at the expense of global object structure and semantics**.

**Analysis of aspect ratio generalization visualization.** As demonstrated in Figs. 10 to 15, NiT-XL demonstrate superior aspect ratio generalization performance than EDM2-L and FlowDCN-XL. Compared to the NiT-XL, both EDM2-L and FlowDCN-XL exhibit several notable shortcomings in image generation quality:

1. *Cropping-Induced Bias.* Long or wide objects, such as guitars (i.e., *class-402*) or parrots (i.e., *class-88,89*), are often truncated or improperly framed. This suggests **a form of information leakage or truncation bias introduced by over-reliance on square or tightly cropped training samples**, leading to unnaturally framed and truncated object boundaries. This corresponds to the finding in SDXL [57]: cropping image data can leak into the generated samples, causing malicious effects, especially in extreme aspect ratios (see Figs. 10 and 11).

2. *Blurring and Lack of Detail.* Many generated images, such as the sea turtle (*class-33*) in FlowDCN-XL or the flowers (*class-985*) in EDM2-L, lack the sharpness and textural richness seen in the NiT-XL outputs. This indicates poor high-frequency detail modeling, which compromises the realism and clarity of the outputs.

3. *Object Repetition and Spatial Artifacts.* There is frequent duplication of objects and structurally incoherent arrangements, breaking spatial consistency. Some entities appear anatomically incorrect or exhibit unnatural part arrangements, especially in animal classes.

4. *Color Artifacts.* Inconsistent coloring and unnatural saturation further diminish the realism of the generated images.

The visualization highlights **a critical limitation of conventional training pipelines that rely on center cropping and resizing to square resolutions**. Models like EDM2-L and FlowDCN-XL, which were trained under such regimes, often fail to generalize to objects with other aspect ratios. This is particularly evident in examples such as guitars (*class-402*), parrots (*class-88,89*), and volcanoes (*class-980*) that are naturally elongated either horizontally or vertically. In contrast, NiT-XL demonstrates robust aspect ratio generalization, preserving the spatial integrity and composition of elongated objects without distortion, demonstrating the effectiveness of its native resolution modeling.

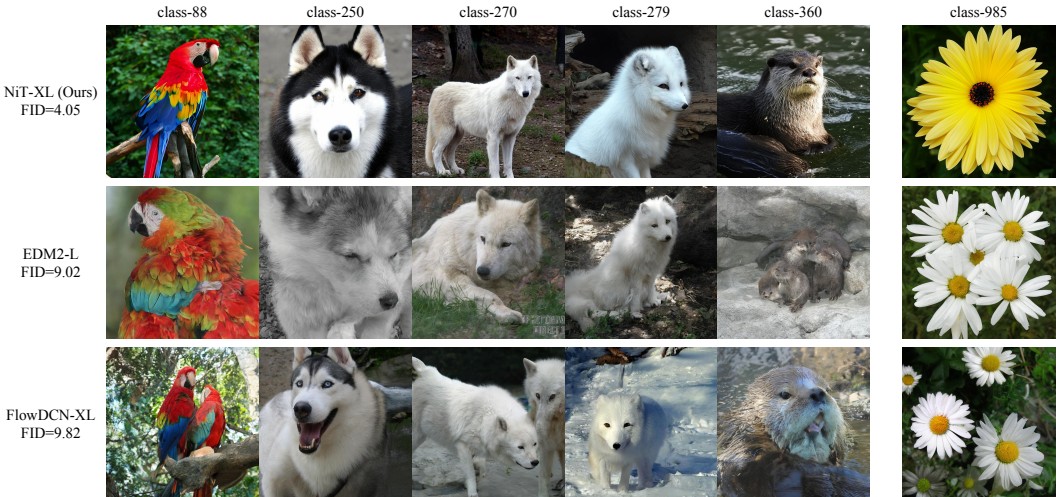

Figure 6: Qualitative comparison of resolution generalization on $768 \times 768$ resolution.

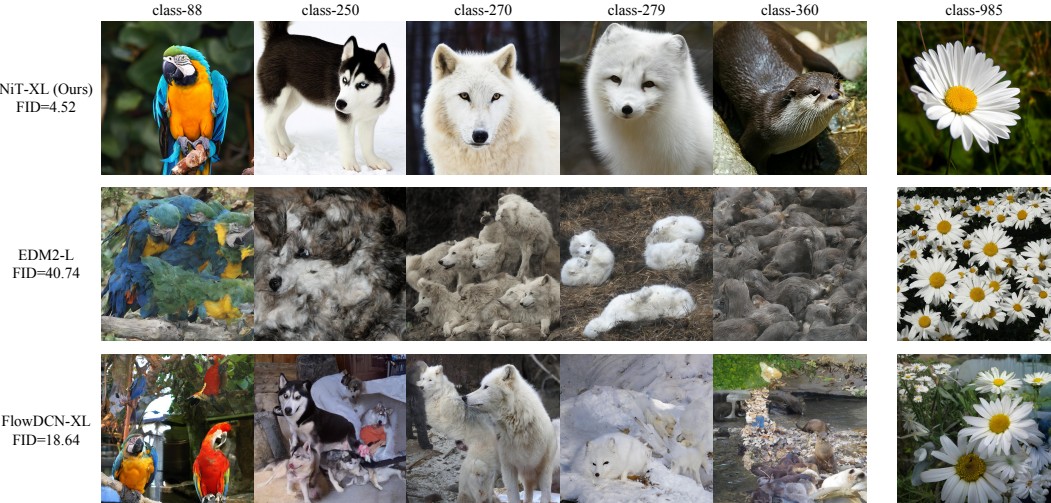

Figure 7: Qualitative comparison of resolution generalization on $1024 \times 1024$ resolution.

## D.2 More Qualitative Results of NiT

More qualititative results of NiT-XL are demonstrated in Figs. 16 to 25

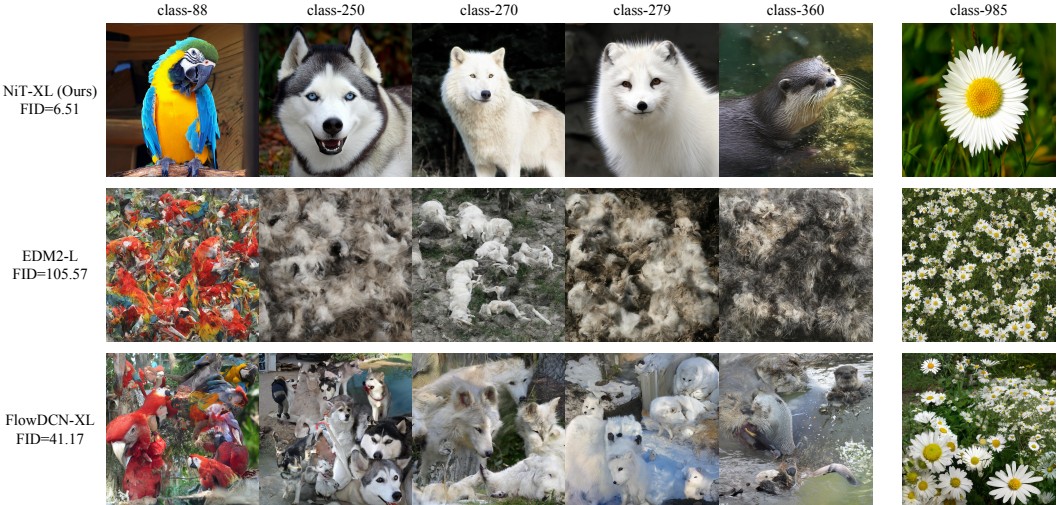

Figure 8: Qualitative comparison of resolution generalization on $1536 \times 1536$ resolution.

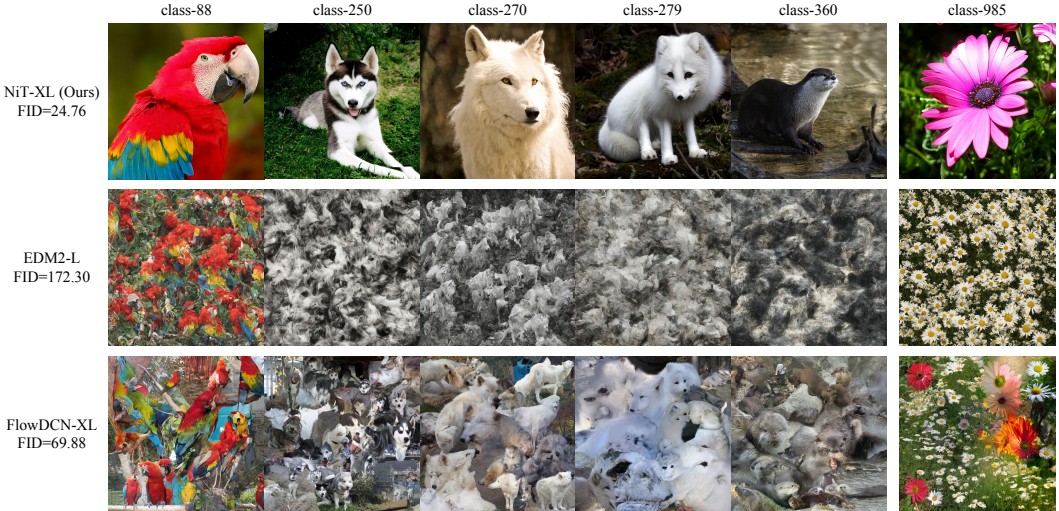

Figure 9: Qualitative comparison of resolution generalization on $2048 \times 2048$ resolution.

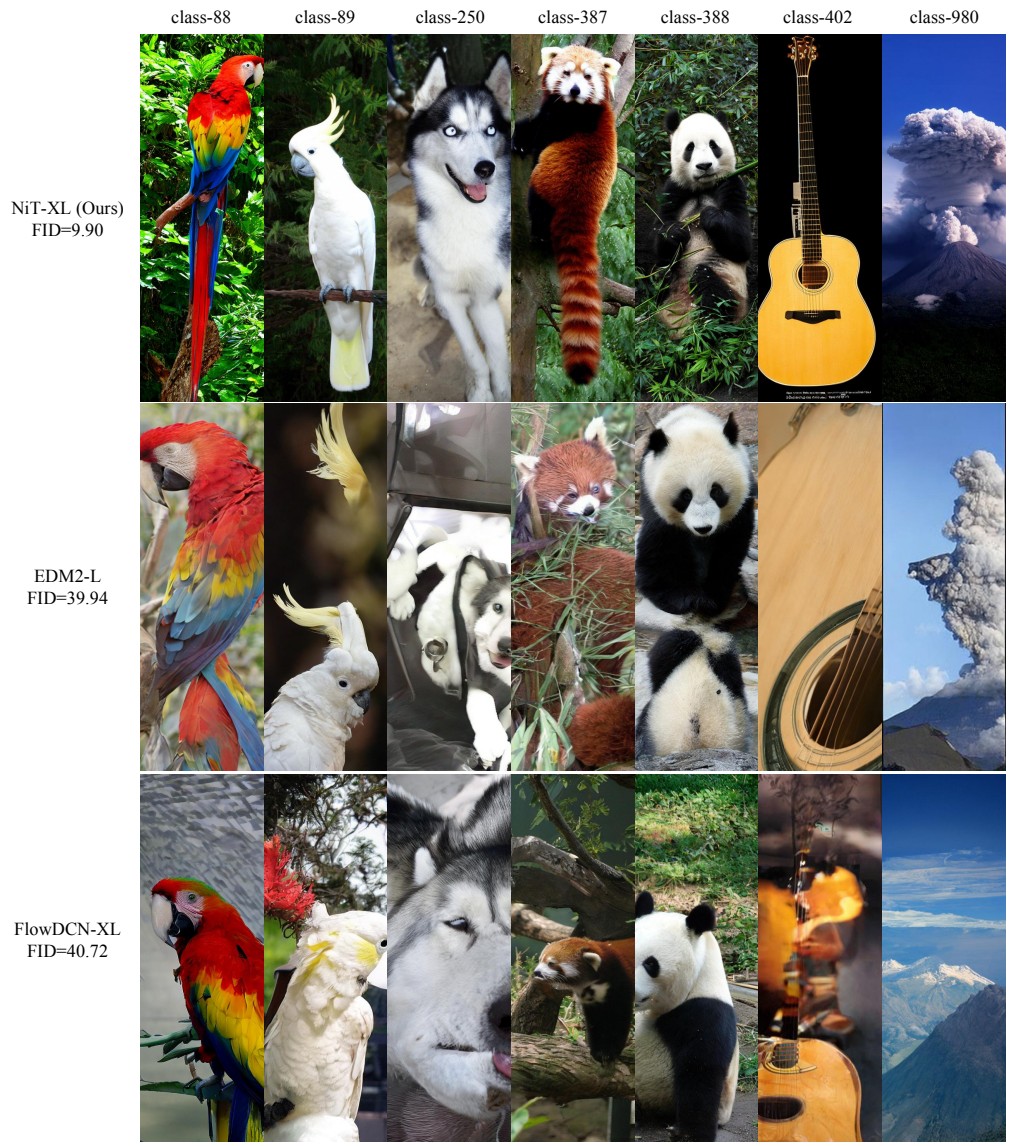

Figure 10: Qualitative comparison of aspect ratio generalization on $960 \times 320$ resolution (corresponding to 3 : 1 aspect ratio).

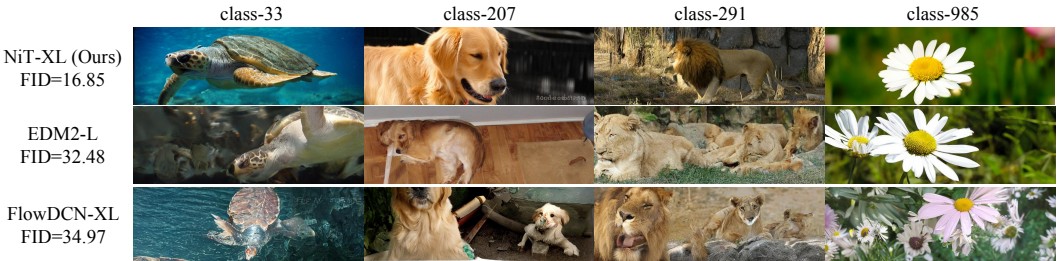

Figure 11: Qualitative comparison of aspect ratio generalization on $320 \times 960$ resolution (corresponding to 1 : 3 aspect ratio).

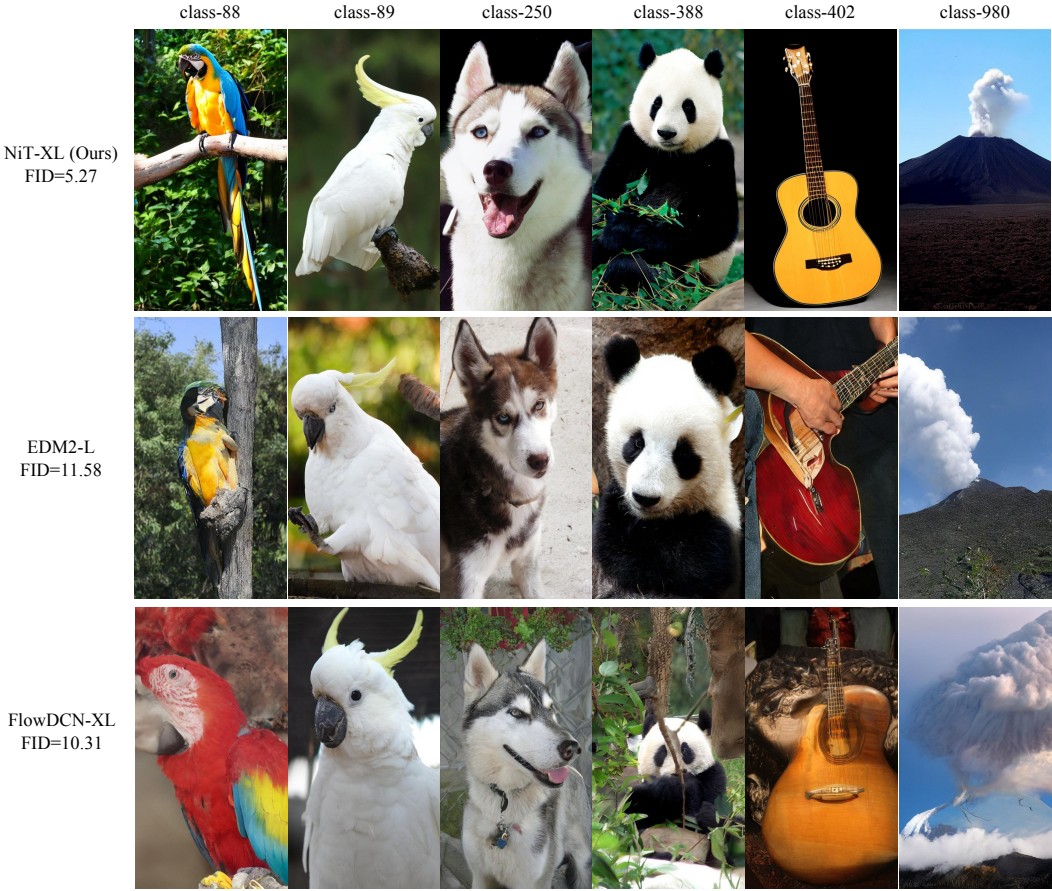

Figure 12: Qualitative comparison of aspect ratio generalization on $768 \times 416$ resolution (corresponding to $16 : 9$ aspect ratio).

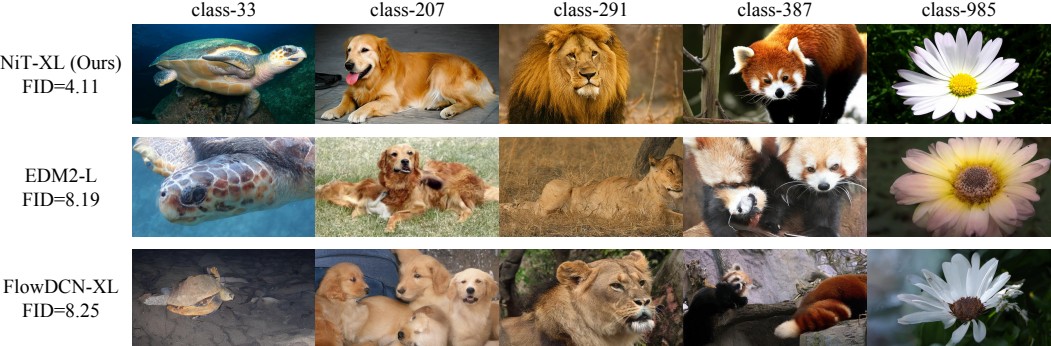

Figure 13: Qualitative comparison of aspect ratio generalization on $416 \times 768$ resolution (corresponding to $9 : 16$ aspect ratio).

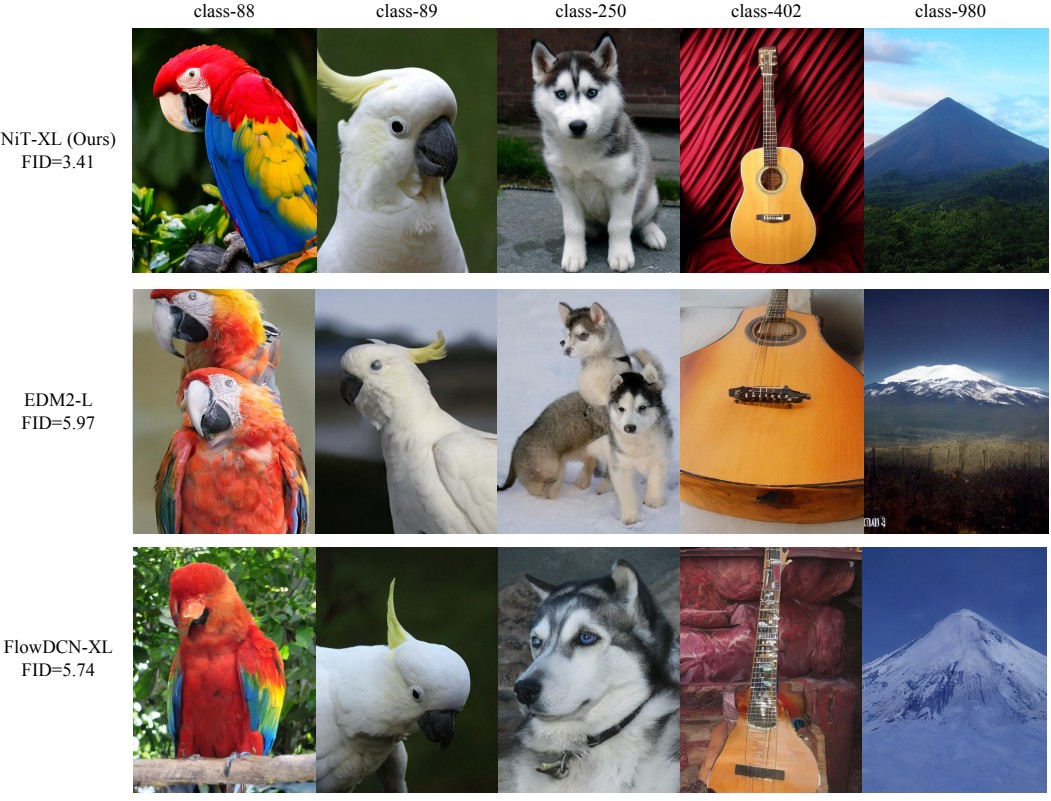

Figure 14: Qualitative comparison of aspect ratio generalization on $640 \times 480$ resolution (corresponding to $4 : 3$ aspect ratio).

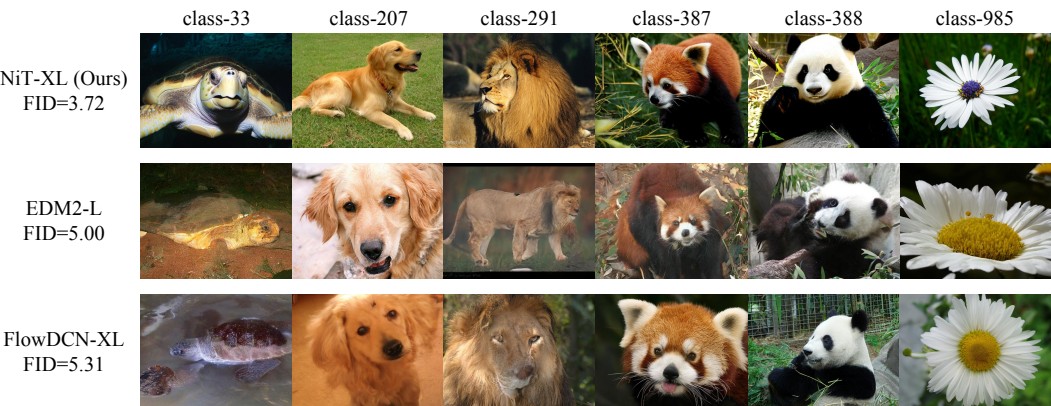

Figure 15: Qualitative comparison of aspect ratio generalization on $480 \times 640$ resolution (corresponding to $3 : 4$ aspect ratio).

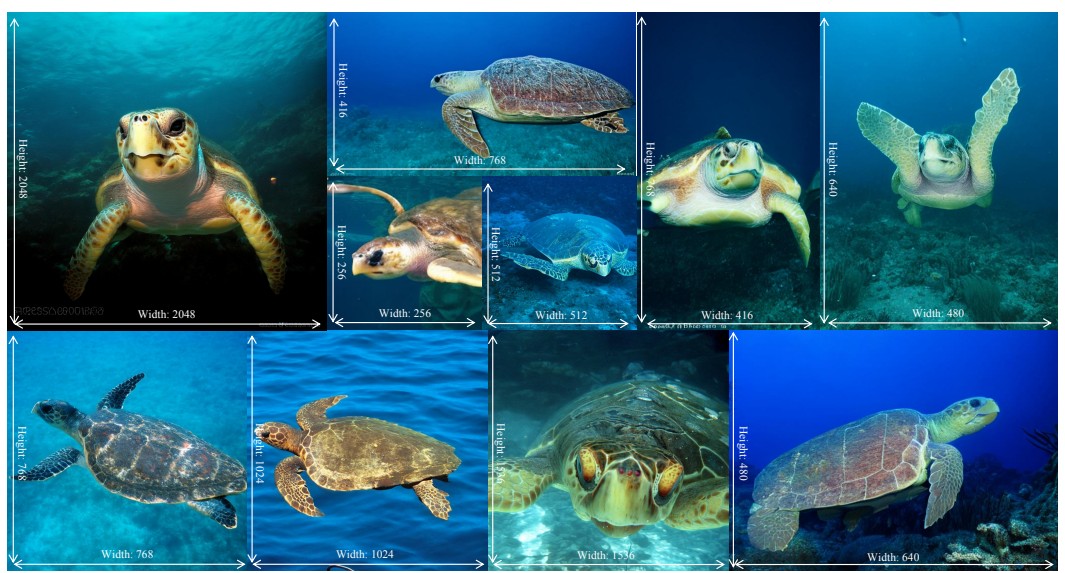

Figure 16: Uncurated generation results of NiT-XL. We use the class label as 33.

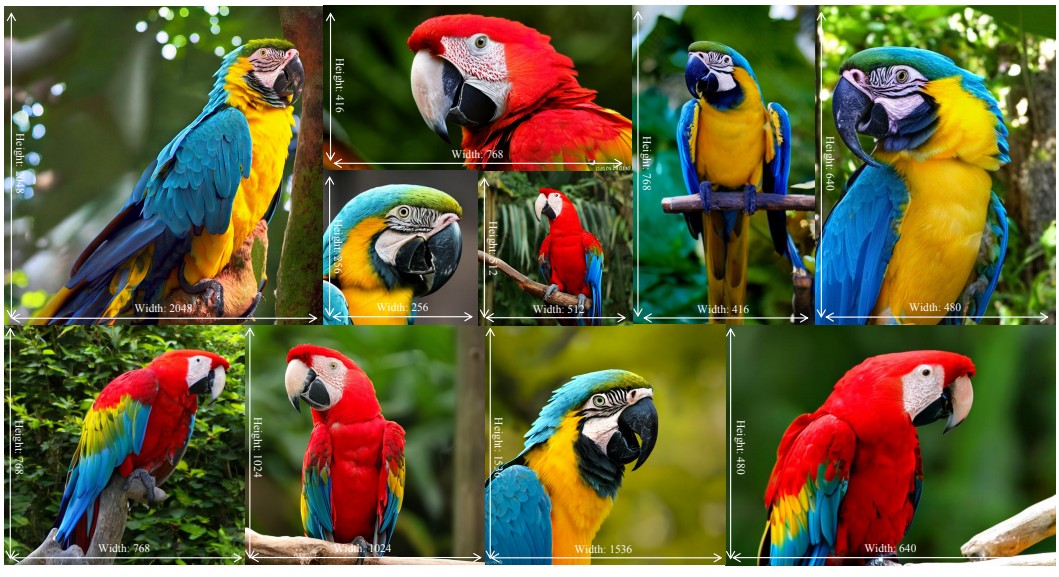

Figure 17: Uncurated generation results of NiT-XL. We use the class label as 88.

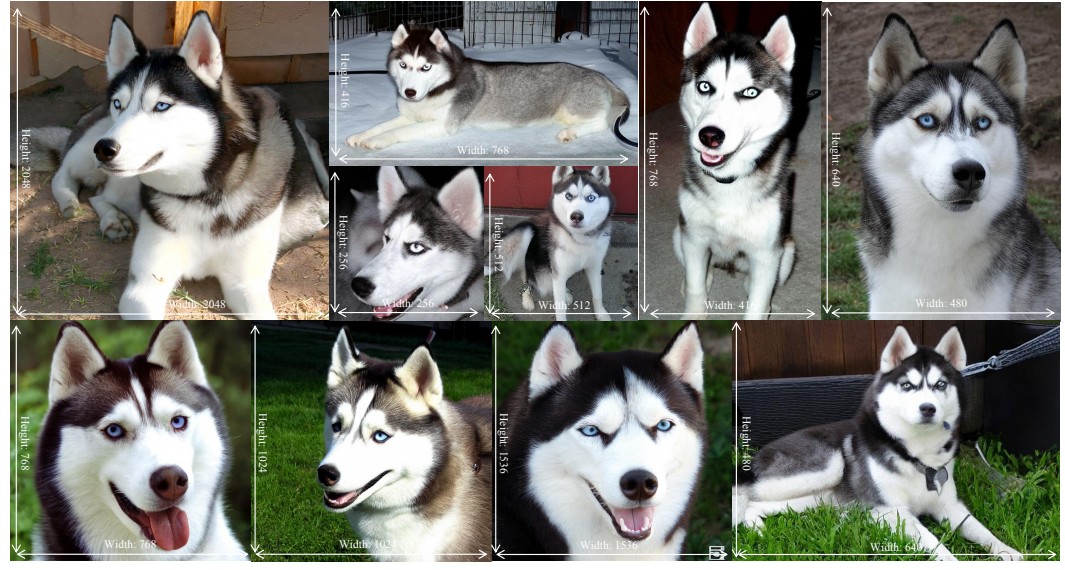

Figure 18: Uncurated generation results of NiT-XL. We use the class label as 250.

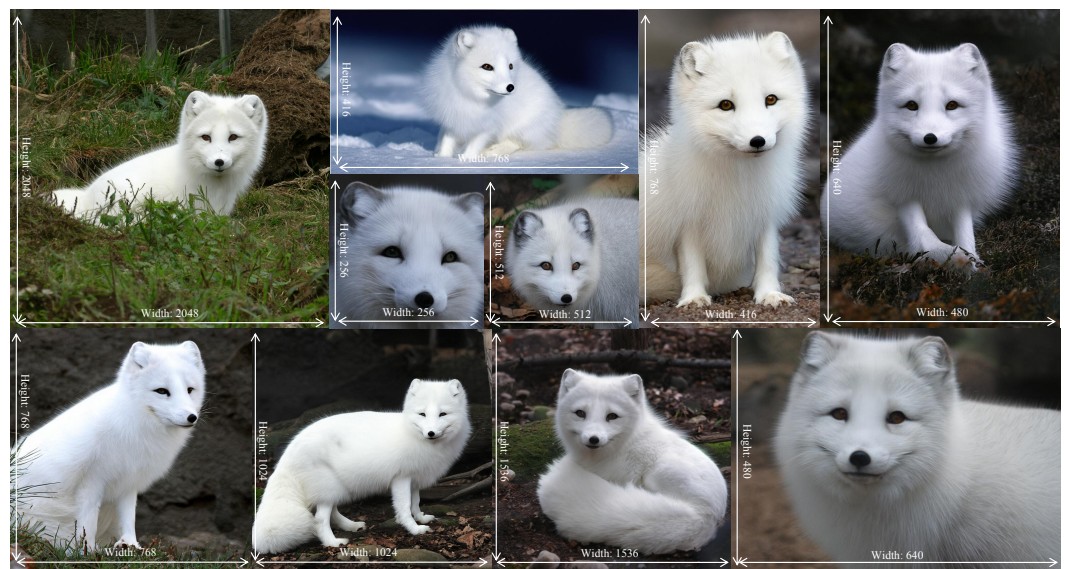

Figure 19: Uncurated generation results of NiT-XL. We use the class label as 279.

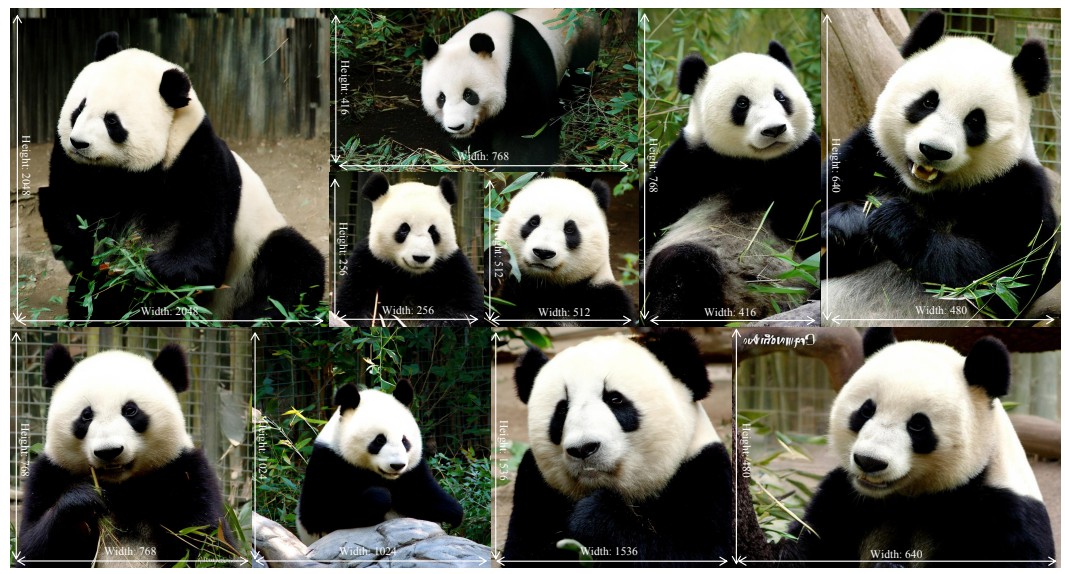

Figure 20: Uncurated generation results of NiT-XL. We use the class label as 388.

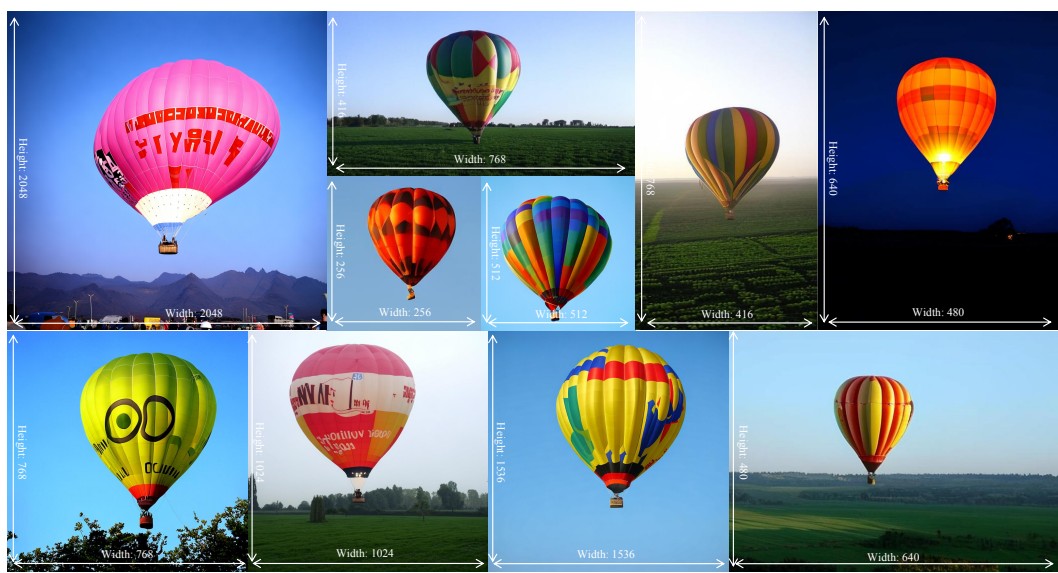

Figure 21: Uncurated generation results of NiT-XL. We use the class label as 417.

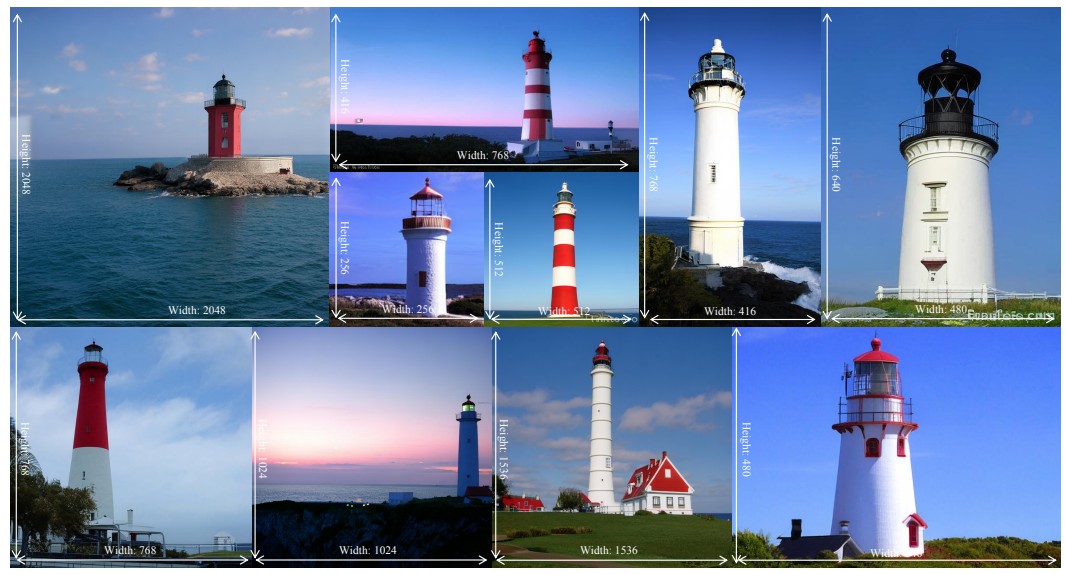

Figure 22: Uncurated generation results of NiT-XL. We use the class label as 437.

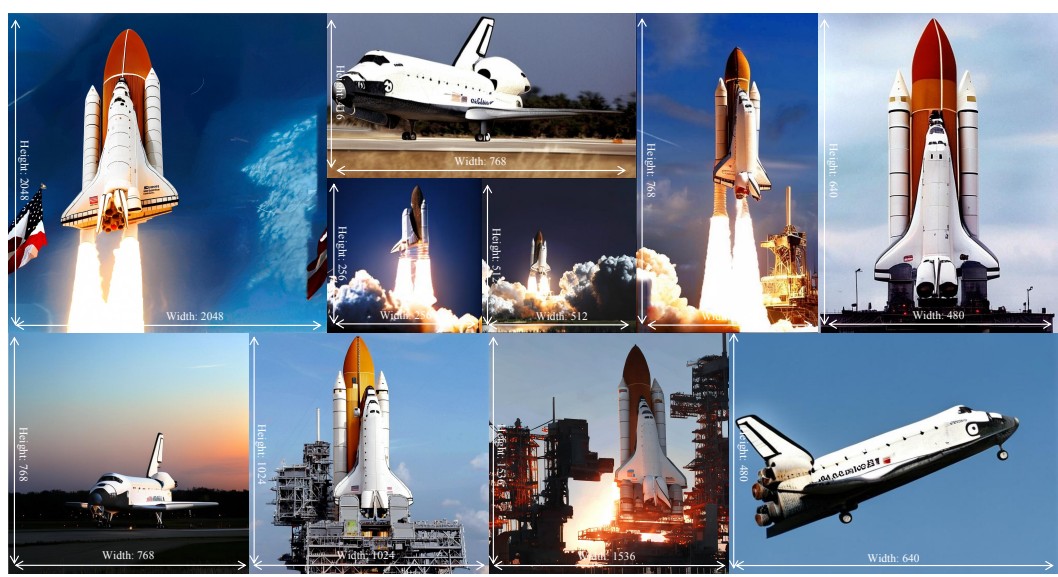

Figure 23: Uncurated generation results of NiT-XL. We use the class label as 812.

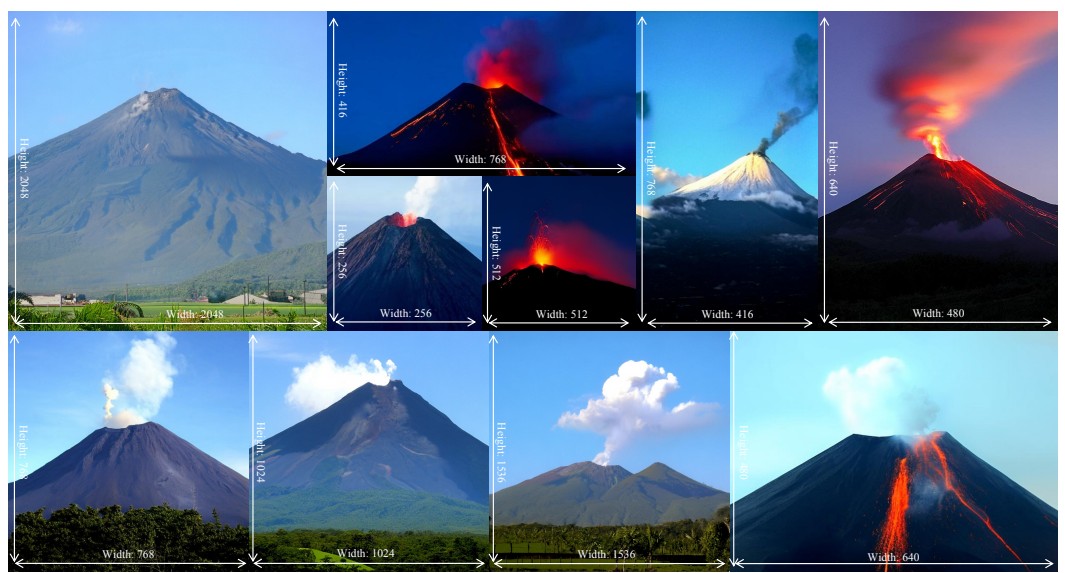

Figure 24: Uncurated generation results of NiT-XL. We use the class label as 980.

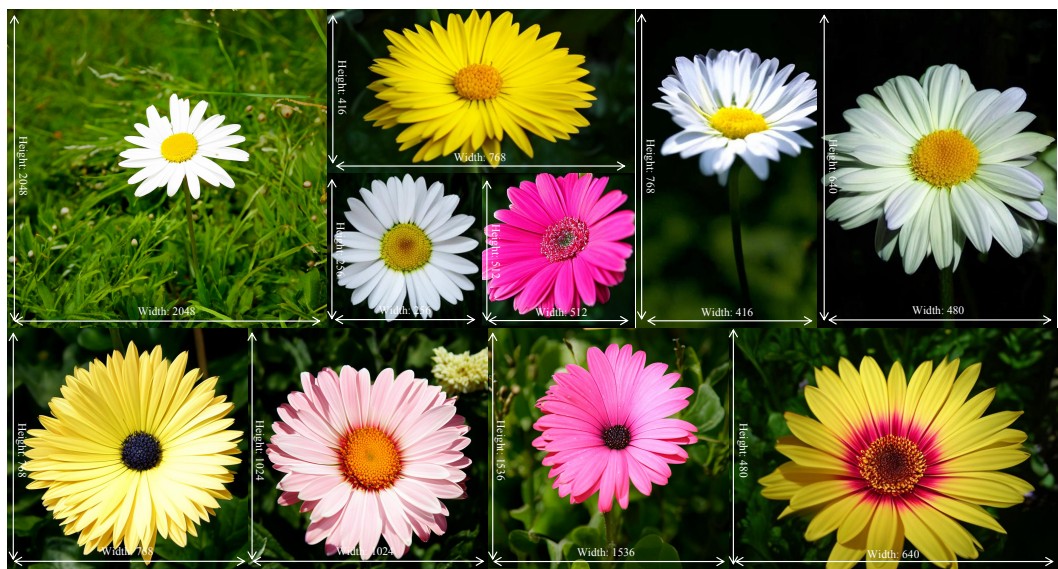

Figure 25: Uncurated generation results of NiT-XL. We use the class label as 985.

