# OpenReview forum: "Native-Resolution Image Synthesis"
_NeurIPS.cc/2025/Conference — NeurIPS 2025 poster_

### Official Review · Reviewer_nzEE · 2025-06-21

**Clarity:** 2
**Significance:** 2
**Originality:** 2
**Rating:** 4
**Confidence:** 3

**Summary:**

This paper introduces Native-resolution image synthesis and proposes the Native-resolution diffusion Transformer (NiT), a generative model that can synthesize images across a wide range of resolutions and aspect ratios without being constrained to a fixed format. Inspired by the variable-length generalization capabilities of LLMs, the model reformulates image generation by directly modeling visual data at its native resolution using techniques like packed variable-length token processing, axial 2D Rotary Positional Embeddings, and FlashAttention. NiT achieves state-of-the-art results on both 256×256 and 512×512 ImageNet benchmarks with a single model and demonstrates strong zero-shot generalization to unseen resolutions and aspect ratios.

**Questions:**

+ Please explain the different between FiT and this work.
+ I think the comparison with other method is not fair. Author should retrain NiT with same VAE as previous method or pick the best previous method (i.e FiT) and train it with DC-AE for a fair comparison.
+ There was a long line of research in native-resolution generation with GAN like ScaleSpace-GAN [1], CREPS [2], AnyRes-GAN [3], INR-GAN [4],.. Author should include them in related work for completeness for future readers/researchers.

Reference:

[1] Wolski et. al. Learning Images Across Scales Using Adversarial Training. SIGGRAPH 2024

[2] Nguyen et. al. Efficient Scale-Invariant Generator with Column-Row Entangled Pixel Synthesis. CVPR 2023

[3] Chai et. al. Any-resolution Training for High-resolution Image Synthesis. ECCV 2022

[4] Skorokhodov et. al. Adversarial Generation of Continuous Images. CVPR 2021

**Ethical Concerns:**

["NO or VERY MINOR ethics concerns only"]

**Final Justification:**

The author address my concerns well but after seeing other concerns from other author, I decide to raise the score to **4** only.

**Limitations:**

No limitations besides then weaknesses I mentioned

**Paper Formatting Concerns:**

no formatting concerns

**Quality:**

3

**Strengths And Weaknesses:**

**Strengths**
+ Impressive generalization on high-resolution compared to previous work, as shown in Tab. 2 and Tab. 3.
+ Theory-wise, this is not new as FiT already explore similar concept of using 2D Rotary Position Embedding (RoPE). **BUT** implementation-wise, this paper propose an very interesting and efficient trick (Packed Full-Attention and Packed Adaptive Layer Normalization) to train diffusion transformer on high-resolution.

**Weaknesses**
+ The different of this work against FiT is not clear, they seem similar.
+ The comparison with other work seem unfair as previous work utilize different VAE than the one used in this work (DC-AE).

---

> ### Author Rebuttal · Authors · 2025-07-30
>
> Thank you sincerely for your valuable suggestions and thorough review. We have thoughtfully addressed each of your points, as detailed in the responses below:
>
> **1. Key distinctions between FiT and NiT**
>
> | Aspect | FiT | NiT |
> |---|---|---|
> | **Training paradigm** | *Padding-and-masking.* FiT only supports a small range of resolutions (i.e., from 64 to 512 on ImageNet). A global maximum token length is fixed; shorter sequences are right-padded and the padding positions are masked during loss computation. For training stability and balance, it only supports resolutions with similar token length. | *Native-resolution packing.* NiT supports training with a large resolution range (i.e., from 32 to 2048 on ImageNet). Latent tokens from multiple images are concatenated until a preset **max sequence length** is reached. Because only real tokens are processed, GPU utilisation and training throughput rise markedly. |
> | **Generalization principle** | Learns mainly at the limited resolution range; out-of-distribution sizes rely on RoPE extrapolation tricks such as NTK/YaRN. | Directly optimises on the natural resolution distribution of the dataset, preserving full spatial hierarchy and learning scale-independent priors. Hence, it natively supports arbitrary resolutions and aspect ratios without auxiliary tricks. |
> | **Architecture** | **Masked full attention**  $$\mathrm{Attn}(Q,K,V)=\text{softmax}\left(\frac{QK^\top}{\sqrt{d_k}}+M\right)V,$$ where $M_{ij}=0$ for image tokens and $M_{ij}=-\infty$ for padding tokens. | **Packed full attention + packed AdaLN.** For a packed sequence whose \(i\)-th image contributes $N_i=\frac{h_i w_i}{p^{2}}$ tokens, cumulative offsets $\texttt{CuSeqLens}=[0,N_1,\dots,\sum_{j=1}^{n}N_j]$ are fed to FlashAttention-2:  $$\mathrm{Attn}(Q,K,V)=\texttt{FlashAttn}(Q,K,V,\texttt{CuSeqLens}).$$  Packed AdaLN uses broadcasting to align conditioning vectors with variable-length image segments. The details of packed attention and packed AdaLN are in section C of the supplementary material.  |
>
> **Summary.** NiT replaces FiT’s padding-centric regime with a packing strategy that eliminates dummy tokens, learns directly on native resolutions, and introduces packed attention and packed AdaLN modules that respect variable sequence boundaries. NiT learns the intrinsic spatial hierarchy of original inputs, capable of generating a more diverse range of resolutions and aspect ratios. This generalization is fundamentally different from FiT's reliance on RoPE extrapolation techniques for generalization.
>
>
> **2. Fair comparison between FiT and NiT**
>
> Both using DC-AE, we train the FiT-B/1 (i.e., 131M model size with the patch size of 1) and NiT-B/1 model with the same training configurations (131,072 token budget with 100K training steps, learning rate 1e-4). We provide the FID of FiT and NiT among diverse resolutions as follows:
> | Method | 256x256 | 512x512 | 768x768 | 480x640 | 432x768 | 320x960 |
> | ------ | ------- | ------- | ------- | ------- | ------- | ------- |
> | FiT    | 41.01 | 26.69  | 152.93 | 57.20 | 84.02  | 124.25 |
> | NiT    | 24.54  | 17.62 | 21.03 | 21.99 | 33.57  | 87.73 |
>
> The above table reveals the superiority of our NiT compared to FiT. NiT surpasses the performance of FiT on both 256x256 and 512x512 benchmarks. Besides, NiT demonstrates more stable and robust resolution generalization capability compared with FiT.
>
>
> **3. Additional related work**
>
> We apologize for the lack of related work on GAN-based native-resolution studies. We sincerely thank the reviewer for the constructive suggestion and would add these papers to the related work in revised manuscript.

---

> ### Comment · Reviewer_nzEE · 2025-08-05
>
> The author address my concerns well but after seeing other concerns from other author, I decide to raise the score to **4** only.

---

### Official Review · Reviewer_5tpv · 2025-06-27

**Clarity:** 4
**Significance:** 3
**Originality:** 3
**Rating:** 5
**Confidence:** 4

**Summary:**

This paper proposes a Native-resolution diffusion Transformer (NiT) model for image synthesis at arbitrary resolutions and aspect ratios. The key contributions are:
1. Introducing the concept of native-resolution image synthesis, which overcomes the limitations of fixed-resolution, square-image methods.
2. Developing the NiT architecture that inherently integrates dynamic tokenization, Variable-Length Sequence Processing and 2D Structural Prior Injection into a single framework.

**Questions:**

1. Please describe the hyperparameters, data preprocessing steps, and exact training procedures in more detail. This will help ensure the reproducibility of your results.
2. Please provide a more in-depth discussion of the limitations of the NiT model, especially regarding its performance on extremely high resolutions and aspect ratios.
3. Please discuss both potential positive and negative societal impacts of your work. A comprehensive discussion of limitations and potential avenues for future improvement.
4. Please include comparisons with more state-of-the-art baseline methods, especially those that are recent and relevant to your work.

**Ethical Concerns:**

["NO or VERY MINOR ethics concerns only"]

**Limitations:**

Providing a more comprehensive discussion of the model's limitations, especially in extreme conditions. On the other hand, while the experiments look very impressive, I need more arguments on the idea of the proposed method, since it is built on some existing techniques.

**Quality:**

3

**Strengths And Weaknesses:**

Strengths:
Quality‌: The proposed NiT model shows impressive performance on both standard and zero-shot generalization tasks, outperforming existing methods. The ablation studies provide insights into the model's ability to generalize.
Clarity‌: The paper is well-written and easy to follow. The motivation, methodology, and experimental setup are clearly explained.
Significance‌: The native-resolution image synthesis paradigm has the potential to bridge the gap between visual generative modeling and large language models, offering a more flexible and scalable approach.
Originality‌: The idea of native-resolution modeling and the NiT architecture are novel.
Weaknesses:
Quality‌: The performance on extremely high resolutions and aspect ratios is still not satisfactory, indicating room for improvement.
Clarity‌: Some details of the implementation, such as hyperparameters and exact training procedures, could be more thoroughly described to ensure reproducibility.
Originality‌: While the main idea is novel, some components of the NiT architecture build upon existing techniques. More arguments are needed.

---

> ### Author Rebuttal · Authors · 2025-07-30
>
> We truly thank the reviewer for your detailed review and thoughtful comments in helping us enhance the quality of our paper. Below, we provide detailed point-to-point responses addressing each of your valuable suggestions:
>
> **1. Detailed experimental setup**
> * Hyper-parameters. As batch size is unsuitable in our packed image processing, we use the token budget (i.e., the summation of total tokens in all training iterations) to represent the training costs. For each iteration, we use $131,072$ tokens, which corresponds to a $512$ batch size on $256\times256$ generation of DiT models. We train our NiT-XL model for $1000K$ steps, with AdamW optimizer and a constant learning rate of $1\times{10}^{-4}$. Mixed-precision training is adopted with $\texttt{torch.bfloat16}$.
> * Data pre-processing. We use DC-AE to convert an image with the shape of $H\times W$ into the latent tokens with the shape of $\frac{H}{32}\times\frac{W}{32}$. All the images are processed without the cropping operation to avoid its negative effects.
> * Training procedure. To support packed training, we pre-counted the number of latent codes corresponding to different images, given a preset maximum length, we flattened the different latent tokens and packed them into one sequence. We use the longest-pack-first histogram packing algorithm to pack the images into different sequences. In our setting, the maximum sequence length is similar to the batch size in normal training. The instance number among different packed sequences is dynamic because diverse image resolutions lead to diverse latent token numbers. In the training stage, we directly train NiT with these packed sequences, as our NiT architecture (packed full-attention and packed AdaLN modules) supports efficient packed processing.
> * Reproducibility. We would open-source all the code, scripts, datasets, and model checkpoints to ensure reproducibility.
>
>
> **2. Limitation**
>
> **2.1 Resolution scalability**
> To further evaluate the resolution scalability of NiT, we study the resolution extrapolation law of NiT. First, we counted the average sequence length of the whole training ImageNet dataset $L_\text{avg}$. We find that NiT takes an average sequence length of $L_{\text{avg}}=171$ during training. Then we calculate the sequence length and the FID score of these extrapolated resolutions, detailed in the following table:
> | resolution          | 512x512 | 768x768 | 1024x1024 | 1280x1280 | 1536x1536 |
> | ------------------- | ------- | ------- | --------- | --------- | --------- |
> | Sequence length $L$   | 256     | 576     | 1024      | 1600      | 2304      |
> | Extrapolation ratio $r$ ($L$/$L_{\text{avg}}$  ) | 1.497   | 3.368   | 5.988     | 9.356     | 13.473    |
> | FID                 | 1.57    | 4.05    | 5.49      | 5.91      | 6.51      |
>
> Thus, we can quantify the resolution scalability of NiT by an extrapolation ratio $r = L/L_{\text{avg}}$, where $L$ is the longer side of the test image and $L_{\text{avg}}$ is the average longer side in the training set. Empirically, we observe a remarkably tight logarithmic relationship between this ratio \$r$ and FID metrics, captured by $\text{FID}(r) = 2.234\cdot\ln r + 1.0224$ with an $R^{2}=0.9641$, indicating that the formula explains more than 96 % of the variance across resolutions. The slope 2.234, therefore, quantifies the resolution scalability of NiT, effectively describing the high-resolution extrapolation quality of NiT. We will incorporate these findings and discuss potential strategies to flatten this slope and further improve high-resolution fidelity in the revised manuscript.
>
> **2.2 Aspect ratio scalability.**  NiT-XL is trained on native-resolution images and therefore transfers smoothly to the common photographic aspect ratios—9 : 16, 3 : 4, 4 : 3, and 16 : 9—without any special handling. The vast majority of ImageNet training images fall within the continuous band [9/16, 16/9], and within this range, the model preserves object shapes and spatial coherence. Once the aspect ratio drifts outside that interval, however, objects start to appear unnaturally stretched or compressed. The most pronounced degradation occurs at the extremes of 1 : 3 and 3 : 1, ratios that are virtually absent from the training distribution. As a result, we treat [1/3, 3] as the current practical upper bound for NiT’s aspect-ratio generalization; supporting ratios beyond this would likely require targeted data augmentation or architectural adjustments.
>
> **2.3 Rationale for the current limitation and future work**
> * The chief bottleneck is the narrow resolution distribution in ImageNet itself. As shown in Fig. 2(a) of the manuscript, most ImageNet images fall between 200 and 600 pixels on their longer side, with very few exceeding 800 pixels. Consequently, the average sequence length is only 171 tokens, providing limited signal for learning truly scale-invariant representations.
> * Looking ahead, we plan to (i) design architectures and training schemes that explicitly target resolution and aspect-ratio robustness, and (ii) study a dedicated “aspect-ratio scaling law” for NiT, which we expect to be more nuanced than the resolution law reported here. These steps should extend the model’s effective operating range beyond the present [1/3, 3] aspect-ratio band and improve high-resolution fidelity.
>
> **3. Societal impacts of this work**
>
> **3.1 Positive societal impacts**
> * Enhanced Image Generation for Broad Applications: By accommodating various resolutions and aspect ratios within a single diffusion model, this approach can streamline workflows across industries (e.g., graphic design, healthcare imaging, education) where high-fidelity images at diverse scales are crucial.
> * Reduced Computational Overhead: Training one unified model rather than multiple resolution-specific models may lower overall computational costs and carbon footprint, benefiting sustainability efforts.
> * Democratization of High-Quality Image Synthesis: The ability to handle images at unseen scales may make advanced generative capabilities more accessible to smaller organizations or researchers with limited resources, fostering innovation.
>
> **3.2 Negative societal impacts**
> * Potential for Misinformation and Deepfakes: As generative models improve in fidelity, they can be misused to create highly convincing fake images or videos, posing risks to social trust and security.
> * Impact on Creative Industries: Widespread adoption of automated image generation tools could disrupt commercial art, photography, or design sectors, raising questions about job displacement and the value of human creativity.
>
>
> **4. Comparisons with more SOTA baselines**
>
> We compare our NiT with more recent SOTA baselines: DiG [1] (CVPR2025), DeepFlow [2], GigaTok [3] (ICCV2025), RandAR [4] (ICML2025), and UCGM [5]. The results are summarized as follows:
> | Method       | Params | 256x256 FID$\downarrow$ | 256x256 sFID$\downarrow$ | 256x256 IS$\uparrow$ | 512x512 FID$\downarrow$ | 512x512 sFID$\downarrow$ | 512x512 IS$\uparrow$ | mFID$\downarrow$  |
> |--------------|--------|-------------|--------------|------------|-------------|--------------|------------|-------|
> | DiG [1]         | 675M   | 2.07        | 4.53         | 278.95     | -           | -            | -          | -     |
> | DeepFlow [2]    | 675M   | 1.77        | 4.44         | 271.3      | 1.96        | 4.28         | 260.3      | 1.87  |
> | GigaTok [3]     | 1.4B   | 2.03        | -            | -          | -           | -            | -          | -     |
> | RandAR [4]      | 1.4B   | 2.15        | -            | 321.97     | -           | -            | -          | -     |
> | UCGM [5]        | 675M   | 2.10        | -            | -          | 2.11        | -            | -          | 2.10  |
> | NiT (Ours)   | 675M   | 2.16        | 6.34         | 253.44     | 1.57        | 4.13         | 260.69     | 1.86  |
>
> Our NiT model demonstrates superior performance compared to these baselines on both ImageNet-$256\times256$ and ImageNet-$512\times$ benchmarks. Notably, we use a single model to compete on the two benchmarks.
>
> *Reference*
>
> [1] Zhu, Lianghui, Zilong Huang, Bencheng Liao, Jun Hao Liew, Hanshu Yan, Jiashi Feng, and Xinggang Wang. "Dig: Scalable and efficient diffusion models with gated linear attention." CVPR, 2025.
>
> [2] Shin, Inkyu, Chenglin Yang, and Liang-Chieh Chen. "Deeply supervised flow-based generative models." arXiv preprint arXiv:2503.14494 (2025).
>
> [3] Xiong, Tianwei, Jun Hao Liew, Zilong Huang, Jiashi Feng, and Xihui Liu. "Gigatok: Scaling visual tokenizers to 3 billion parameters for autoregressive image generation." arXiv preprint arXiv:2504.08736 (2025).
>
> [4] Pang, Ziqi, Tianyuan Zhang, Fujun Luan, Yunze Man, Hao Tan, Kai Zhang, William T. Freeman, and Yu-Xiong Wang. "Randar: Decoder-only autoregressive visual generation in random orders." CVPR, 2025.
>
> [5] Sun, Peng, Yi Jiang, and Tao Lin. "Unified Continuous Generative Models." arXiv preprint arXiv:2505.07447 (2025).

---

### Official Review · Reviewer_dH7i · 2025-07-01

**Clarity:** 4
**Significance:** 3
**Originality:** 4
**Rating:** 4
**Confidence:** 3

**Summary:**

This paper introduces "native-resolution image synthesis," a new paradigm for generative modeling that handles images of arbitrary resolutions and aspect ratios. The authors propose the Native-resolution diffusion Transformer (NiT), which is inspired by Large Language Models and processes variable-length visual tokens without resizing or cropping inputs. The proposed model is shown to achieve state-of-the-art results on standard benchmarks and demonstrates impressive zero-shot generalization to unseen resolutions and aspect ratios.

**Questions:**

I like this paper which would benefit to the community in my opinion.

**Ethical Concerns:**

["NO or VERY MINOR ethics concerns only"]

**Final Justification:**

Thanks for the authors' reply. Most of my concerns are solved. Considering the comments from other reviewers, I would keep my rating.

**Limitations:**

yes

**Quality:**

3

**Strengths And Weaknesses:**

Strengths:
1. The paper addresses a significant and practical limitation of current generative models, which are typically constrained to fixed, square resolutions. The proposed native-resolution approach is a novel and impactful contribution to the field.
2. The technical design of the NiT architecture is sound and well-motivated, cleverly adapting concepts like variable-length sequence processing and rotary positional embeddings (2D RoPE) from LLMs to the vision domain.

Weaknesses:
1. The evaluation relies heavily on quantitative metrics. The paper would benefit from a more extensive qualitative analysis in the main text, particularly showing failure cases to better illustrate where the model struggles with object coherence or detail at unseen scales.
2. While the generalization is strong, the model's performance degrades noticeably at extremely high resolutions like 2048x2048 and on extreme aspect ratios. A deeper analysis of these failure points would provide a more complete understanding of the method's limitations.
3. The ablation study confirms the importance of native-resolution training data but could further explore how the specific distribution of resolutions and aspect ratios within the training set impacts generalization performance.

---

> ### Author Rebuttal · Authors · 2025-07-30
>
> Thank you very much for your insightful feedback and careful review, which has greatly helped improve the manuscript. In response to your comments, we have carefully prepared point-to-point clarifications as follows:
>
> **1. Qualitative analysis**
> * We thank the reviewer for suggesting more qualitative results in the main text, and we will add a detailed qualitative analysis in the following revision. Currently, we have provided the qualitative results of our model and the comparison with baselines EDM and FlowDCN in the supplementary material.
> * We regret that our submission did not include visual examples of failure cases. In follow-up tests at very high resolutions (e.g., 2048 × 2048), we indeed observed spatial artefacts: certain objects are rendered with anatomically implausible features or with parts arranged unnaturally, compromising spatial coherence. Because rebuttal guidelines prohibit anonymous external links, we cannot attach the images here; however, we will include representative visualizations, together with an analysis of their causes and possible mitigations, in the revised manuscript.
>
>
> **2. The limitation of this method**
>
> **2.1 Resolution scalability**
> To further evaluate the resolution scalability of NiT, we study the resolution extrapolation law of NiT. First, we counted the average sequence length of the whole training ImageNet dataset $L_\text{avg}$. We find that NiT takes an average sequence length of $L_{\text{avg}}=171$ during training. Then we calculate the sequence length and the FID score of these extrapolated resolutions, detailed in the following table:
> | resolution          | 512x512 | 768x768 | 1024x1024 | 1280x1280 | 1536x1536 |
> | ------------------- | ------- | ------- | --------- | --------- | --------- |
> | Sequence length $L$   | 256     | 576     | 1024      | 1600      | 2304      |
> | Extrapolation ratio $r$ ($L$/$L_{\text{avg}}$}  ) | 1.497   | 3.368   | 5.988     | 9.356     | 13.473    |
> | FID                 | 1.57    | 4.05    | 5.49      | 5.91      | 6.51      |
>
> Thus, we can quantify the resolution scalability of NiT by an extrapolation ratio $r = L/L_{\text{avg}}$, where $L$ is the longer side of the test image and $L_{\text{avg}}$ is the average longer side in the training set. Empirically, we observe a remarkably tight logarithmic relationship between this ratio \$r$ and FID metrics, captured by $\text{FID}(r) = 2.234\cdot\ln r + 1.0224$ with an $R^{2}=0.9641$, indicating that the formula explains more than 96 % of the variance across resolutions. The slope 2.234, therefore, quantifies the resolution scalability of NiT, effectively describing the high-resolution extrapolation quality of NiT. We will incorporate these findings and discuss potential strategies to flatten this slope and further improve high-resolution fidelity in the revised manuscript.
>
> **2.2 Aspect ratio scalability.**  NiT-XL is trained on native-resolution images and therefore transfers smoothly to the common photographic aspect ratios—9 : 16, 3 : 4, 4 : 3, and 16 : 9—without any special handling. The vast majority of ImageNet training images fall within the continuous band [9/16, 16/9], and within this range, the model preserves object shapes and spatial coherence. Once the aspect ratio drifts outside that interval, however, objects start to appear unnaturally stretched or compressed. The most pronounced degradation occurs at the extremes of 1 : 3 and 3 : 1, ratios that are virtually absent from the training distribution. As a result, we treat [1/3, 3] as the current practical upper bound for NiT’s aspect-ratio generalization; supporting ratios beyond this would likely require targeted data augmentation or architectural adjustments.
>
> **2.3 Rationale for the current limitation and future work**
> * The chief bottleneck is the narrow resolution distribution in ImageNet itself. As shown in Fig. 2(a) of the manuscript, most ImageNet images fall between 200 and 600 pixels on their longer side, with very few exceeding 800 pixels. Consequently, the average sequence length is only 171 tokens, providing limited signal for learning truly scale-invariant representations.
> * Looking ahead, we plan to (i) design architectures and training schemes that explicitly target resolution and aspect-ratio robustness, and (ii) study a dedicated “aspect-ratio scaling law” for NiT, which we expect to be more nuanced than the resolution law reported here. These steps should extend the model’s effective operating range beyond the present [1/3, 3] aspect-ratio band and improve high-resolution fidelity.
>
>
> **3. Impact of specific resolution and aspect-ratio distributions**
>
> We conduct 4 groups of experiments with different dataset distributions to explore this effect. (a) we keep the aspect ratio of image data and resize its height and width smaller than $256$; (b) we keep the aspect ratio of image data and resize its height and width smaller than $512$; (c) we keep the aspect ratio of image data and resize its height and width smaller than $768$; (d) we use native-resolution image data for training without resizing them under certain resolution. We report the FID of each setting and resolution as in the following table.
> | Config | Resolution | 256x256 | 512x512 | 768x768 | 1024x1024 |
> | ------ | ---------- | ------- | ------- | ------- | --------- |
> | (a)    | Below256   | 21.08  | 180.27 | 241.09  | 264.94    |
> | (b)    | Below512   | 36.10 | 28.06  | 195.99 | 285.89    |
> | (c)    | Below768   | 38.89  | 31.64 | 32.63  | 124.14    |
> | (d)    | Full Data  | 31.95  | 24.52  | 25.68  | 32.32    |
>
> We find that the training resolution distribution significantly impacts the generalization ability. When training images are capped at $256$ px on the long side (Config a), the model excels at $256 \times 256$ but degrades sharply at $512 \times 512$ and $768 \times 768$. Raising the cap to $512$ px (Config b) or $768$ px (Config c) postpones—but does not eliminate—the drop-off: once the test resolution exceeds the highest resolution encountered in training, FID climbs rapidly. By contrast, training at native image resolution with no upper bound (Config d) yields consistently strong performance across the full range of resolutions we evaluated. These results underline a simple principle: the broader the resolution spectrum in the training set, the better NiT generalizes, highlighting the necessity of resolution diversity.

---

> > ### Comment · Reviewer_dH7i · 2025-08-09
> > **response**
> >
> > Thanks for the authors' reply. Most of my concerns are solved. Considering the comments from other reviewers, I would keep my rating.

---

### Official Review · Reviewer_S4nj · 2025-07-01

**Clarity:** 3
**Significance:** 2
**Originality:** 2
**Rating:** 4
**Confidence:** 4

**Summary:**

This paper proposes Native-resolution Image Synthesis, a new approach to image generation. It introduces the Native-resolution diffusion Transformer (NiT), which handles images of any resolution or aspect ratio. NiT uses variable-length visual tokens to overcome the limitations of fixed-size models. Trained only on ImageNet, it achieves state-of-the-art results on both 256×256 and 512×512 benchmarks. It also generalizes well to unseen sizes like 1024×1024 and unusual aspect ratios.

**Questions:**

See weakness

**Ethical Concerns:**

["NO or VERY MINOR ethics concerns only"]

**Final Justification:**

weak accept

**Limitations:**

yes

**Quality:**

3

**Strengths And Weaknesses:**

**Strengths**
1. This paper effectively addresses the challenge of using images with diverse resolutions during training. Unlike previous methods that rely on fixed-size inputs, the proposed NiT model learns directly from native-resolution images. As a result, it achieves strong and stable performance not only on standard benchmarks like 256×256 and 512×512, but also on unseen higher resolutions and aspect ratios.

2. The methodology is clearly presented, with well-structured explanations of each component.


**Weakness**
1.The paper briefly mentions multi-resolution training in related work, which has been proven effective and practical in large-scale settings. Its approach is especially similar to the frame packing strategy used in CogVideoX. However, whether the proposed method retains its efficiency and effectiveness at large scale remains an open question, as the paper does not provide empirical validation under such conditions.

2. The paper mentions training with images of diverse resolutions, but lacks concrete details on the experimental setup. For example, it does not specify how image resolutions are mixed within a batch or across GPUs during training. Compared to bucket sampling, it's unclear whether the proposed method can truly support more diverse and continuous resolution ranges in practice. Additionally, since the model relies on a high compression ratio DC-AE with a downsampling factor of 32, all image resolutions must be multiples of 32. This constraint may limit the flexibility and granularity of supported resolutions in real-world applications.

3. The paper emphasizes its ability to resolution extrapolate. However, combining multi-resolution training with RoPE-style positional encoding already enables some level of resolution extrapolation. While the paper presents strong results in this area, it would be helpful to clarify how much of the performance gain comes from the proposed architectural design versus the known benefits of RoPE and resolution diversity in training.

---

> ### Author Rebuttal · Authors · 2025-07-30
>
> We sincerely appreciate your constructive comments and efforts in helping us enhance the quality of our paper. Below, we provide detailed point-to-point responses addressing each of your valuable suggestions:
>
> **1. Effectiveness in large-scale settings**
>
> NiT remains both efficient and effective, even in large-scale text-to-image scenarios. As mentioned in line 222 of our manuscript, we have included detailed setup information and experimental results for the text-to-image generation task in the supplementary materials. Specifically, we trained a model containing $673M$ parameters on the SAM [1] dataset (comprising $10M$ images), utilizing descriptive captions generated by MiniCPM-V. Importantly, NiT-T2I achieves a strong CLIP score of $0.345$ and notably outperforms SD-v1.5 and SDXL-Turbo in both FID and CLIP scores, despite having a significantly smaller model size and lower training costs.
>
> **2. Training with diverse resolutions**
>
> **2.1 Concrete details on experimental setup.**
> We provide the details on multi-GPU parallel in our native-resolution training setting.
> * Preparing a packed distributed sampler. We first construct the metafile, restoring the index and information of each image instance. Given a preset maximum length, we pack different image instances into one sequence, using the longest-pack-first histogram packing algorithm [2]. We record the index of each image instance in a sequence. Then, we use these indices to load the corresponding image instance and pack them into a sequence for each training iteration.
> For example, we first obtain the meta file of each image file with a piece as follows:
>     ```
>     {"index": 0, "image_file": "n01601694/n01601694_11629.JPEG", "image_h": 384, "image_w": 576,}
>     {"index": 1, "image_file": "n01601694/n01601694_11799.JPEG", "image_h": 320, "image_w": 480}
>       ......
>     {"index": 1281166, "image_file":......}
>     ```
> After setting the index of each image instance, we build a packed distributed sampler:
>     ```
>     [
>         [846047, 838285, 837472, 836430, 828275, 807897],
>         [816704, 796202, 793598, 792986, 781934, 769695],
>         .......,
>     ]
>     ```
> Each list records the indices of image instances to be packed into one sequence. Note that the list may have different numbers of image instances, as we adhere to the native token number of each image instance.
> * Given $N$ GPUs, with the above packed distributed sampler, we load the processed image latent codes according to the index provided by the sampler. Then we pack these codes into one sequence independently on each device. On the local devices, the model uses these packed sequences to calculate and then gather the loss to calculate the gradient. Then the gradient back-propagates to update the model parameters.
>
>
> **2.2 Comparison with bucket sampling.**
> The comparison of bucket sampling and native-resolution strategy can be summarized as follows:
> | Aspect                  | Bucket Sampling                          | Native-Resolution Strategy               |
> |-------------------------|------------------------------------------|------------------------------------------|
> | Resolution Handling     | Predefines fixed resolution buckets      | Utilizes dataset's inherent resolution diversity |
> | Image Processing        | Resizes and crops images to fit buckets  | Resizes images to the nearest multiple of 32 |
> | Flexibility             | Limited to discrete set of resolutions   | Integer-continuous resolution range in latent space |
> | Performance Beyond Range| Significant drop in generation quality   | Maintains performance across diverse resolutions |
> | Latent Code Resolution  | Discrete steps based on predefined buckets | Continuous resolution scaling (multiple of 32) |
> | Dataset Utilization     | Constrained by bucket definitions        | Leverages full resolution diversity of training data |
>
> To further validate our claims, we use the same dataset (SAM [1] and JourneyDB [3]) as PixArt-$\alpha$ [4] and train our NiT-XL model to compare the effectiveness of our native-resolution strategy. PixArt-$\alpha$ divides the image size into $40$ buckets with different aspect ratios, each with varying aspect ratios ranging from $0.25$ to $4$. In contrast, our NiT directly learns the native-resolution of these image datasets. We use FID and CLIP Score on the COCO-30K [5] benchmark to evaluate the models. The results are summarized as follows:
>
> | Method | Params | 512x512 FID$\downarrow$ | 512x512 CLIP$\uparrow$ | 768x768 FID$\downarrow$ | 768x768 CLIP$\uparrow$ | 1024x1024 FID$\downarrow$ | 1024x1024 CLIP$\uparrow$ |
> |--------------|--------|---------------|---------------|---------------|---------------|---------------|---------------|
> | PixArt-$\alpha$ | 0.6B   | 315.66  | 0.0461  | 22.42  | 0.3462  | 20.11  | 0.3581  |
> | NiT (Ours)     | 0.67B  | 19.13  | 0.3746  | 5.81  | 0.3592  | 11.95  | 0.3721  |
>
> NiT demonstrates more stable performance on a diverse resolution range than PixArt-$\alpha$. On $512\times512$ resolution, PixArt-$\alpha$ fails to generate meaningful samples, but NiT can generalize to this resolution. Besides, NiT demonstrates superior FID and CLIP Score on other resolutions, validating the effectiveness of our native-resolution training strategy.
>
> **2.3 Reliance on VAE.** We follow the dominant latent diffusion model (LDM) paradigm to train our models, necessitating the usage of an image VAE. We use DC-AE with a $32$ downsampling scale to support super high-resolution (up to $2048$ resolution) image synthesis. For $2048\times2048$ resolution, it leads to $4096$ tokens with DC-AE and $65536$ tokens with SD-VAE [6], which has an $8$ downsampling ratio. In the real-world scenario, the former setting is more practical as it produces fewer tokens, facilitating the device requirement. We recognize that a $32$ downsampling scale may limit the granularity of resolutions, but it is compatible with most normal resolutions (such as $256, 512, 1024, 2048$, etc.) and is more feasible in super high-resolution generation scenarios.
>
>
> **3. Performance gain**
>
> We rigorously conduct experiments to validate the effects of each component. We conduct four groups for this component analysis: (a) we use fixed $512\times512$-resolution data and absolute positional embedding (sin-cos PE) as our positional embedding, corresponding to the SiT [7] baseline. (b) We use fixed $512\times512$-resolution data and RoPE. (c) We use native-resolution data and absolute positional embedding. (d) We use native-resolution data and RoPE, corresponding to our NiT. The following table shows the FID score of each setting and resolution.
> | setting | Resolution | PosEmbed | 256x256 | 512x512 | 768x768 |
> | ------- | ---------- | -------- | ------- | ------- | ------- |
> | (a)     | fxied-512  | AbsPE    | 125.40 | 29.40  | 124.51 |
> | (b)     | fxied-512  | rope     | 127.47 | 24.64  | 86.36  |
> | (c)     | native     | AbsPE    | 35.66  | 29.41 | 31.06 |
> | (d)     | native     | rope     | 24.54  | 17.62 | 21.03 |
>
> The above table demonstrates the performance gain of each component. (1) NiT' s strong generalization to unseen resolutions and aspect ratios is primarily enabled by the training resolution diversity. (a) and (b) learn to overfit the fixed-resolution $512\times512$, thus losing the capability to generalize to other resolutions.  When trained with native-resolution data, both using absolute PE (Config (c)) and using RoPE (Config (d)) can demonstrate strong generalization capability across multiple resolutions. (2) RoPE can accelerate the convergence speed compared to absolute PE. For both fixed-resolution settings and native-resolution settings, RoPE can improve model performance across multiple resolutions.
>
>
>
> *Reference*
>
> [1] Kirillov, Alexander, Eric Mintun, Nikhila Ravi, Hanzi Mao, Chloe Rolland, Laura Gustafson, Tete Xiao et al. "Segment anything." In Proceedings of the IEEE/CVF international conference on computer vision, pp. 4015-4026. 2023.
>
> [2] Krell, Mario Michael, Matej Kosec, Sergio P. Perez, and Andrew Fitzgibbon. "Efficient sequence packing without cross-contamination: Accelerating large language models without impacting performance." arXiv preprint arXiv:2107.02027 (2021).
>
> [3] Sun, Keqiang, Junting Pan, Yuying Ge, Hao Li, Haodong Duan, Xiaoshi Wu, Renrui Zhang et al. "Journeydb: A benchmark for generative image understanding." Advances in neural information processing systems 36 (2023): 49659-49678.
>
> [4] Chen, Junsong, Jincheng Yu, Chongjian Ge, Lewei Yao, Enze Xie, Yue Wu, Zhongdao Wang et al. "Pixart-$\alpha $: Fast training of diffusion transformer for photorealistic text-to-image synthesis." arXiv preprint arXiv:2310.00426 (2023).
>
> [5] Tsung-Yi Lin, Michael Maire, Serge Belongie, James Hays, Pietro Perona, Deva Ramanan, Piotr Dollár, and C Lawrence Zitnick. Microsoft coco: Common objects in context. In ECCV, 2014.
>
> [6] Rombach, Robin, Andreas Blattmann, Dominik Lorenz, Patrick Esser, and Björn Ommer. "High-resolution image synthesis with latent diffusion models." In Proceedings of the IEEE/CVF conference on computer vision and pattern recognition, pp. 10684-10695. 2022.
>
> [7] anye Ma, Mark Goldstein, Michael S Albergo, Nicholas M Boffi, Eric Vanden-Eijnden, and Saining Xie. Sit: Exploring flow and diffusion-based generative models with scalable interpolant transformers. arXiv preprint arXiv:2401.08740, 2024.

---

> > ### Comment · Reviewer_S4nj · 2025-08-07
> > **response**
> >
> > my concerns are well addressed. I will raise my score to 4

---

### Author Response · Authors · 2025-08-08

Dear Reviewers,

Thank you again for your detailed and insightful feedback on our manuscript. We have submitted our rebuttal and hope it has clarified the points you raised.

As the discussion period is nearing its end, we wanted to briefly follow up. We are fully committed to improving our work and would be grateful for the opportunity to address any further questions or remaining concerns you might have.

We deeply appreciate your time and engagement in this process.

Sincerely,
The Authors

---

### Decision · Program_Chairs · 2025-09-17

**Decision:**

Accept (poster)

**Comment:**

This work presents NiT, a Native-resolution Diffusion Transformer that explicitly models varying image resolutions and aspect ratios. Building on top of Diffusion Transformers, the authors propose several improvements, including dynamic tokenization (via longest-pack-first histogram packing algorithm [32]), variable-length sequence processing (via FlashAttention-2), and axial 2D RoPE. A single NiT demonstrates promising results on both ImageNet-256 and ImageNet-512 benchmarks, and shows zero-shot generalization to unseen resolutions and aspect ratios.

Initially, the reviewers raised several concerns, which are briefly outlined below:

* Reviewer S4nj: Transferability to large-scale settings, implementation details, ablation on performance gain.

* Reviewer dH7i: Lack of qualitative analysis, analysis on the failure case for large resolutions, performance generalizability.

* Reviewer 5tpv: Weak performance on large resolutions, implementation details, limited novelty.

* Reviewer nzEE: Similarity to prior work FiT, unfair comparisons with prior works.

The rebuttal and subsequent author-reviewer discussions effectively addressed most of the reviewers' concerns. After carefully considering the reviews, rebuttal, and discussion, the AC concurs with the reviewers’ assessment and thus recommends acceptance of the paper.

While recognizing the engineering contributions (e.g., NiT's adoption of longest-pack-first histogram packing algorithm [23] and FlashAttention-2 [13]), the AC notes that axial-RoPE has been proposed in prior work (e.g., FiT or [A]). A proper discussion and citation of those related works would strengthen the revision. Finally, the authors are encouraged to incorporate the rebuttal experiments into the manuscript and address the reviewers’ feedback in the final revision.

[A] Rotary Position Embedding for Vision Transformer. ECCV 2024.